# Real-time in situ magnetization reprogramming for soft robotics

Xianqiang Bao[1,2,3,10], Fan Wang[1,4,10], Jianhua Zhang[1,5,6,10], Mingtong Li[1], Shuaizhong Zhang[1,7], Ziyu Ren[1], Jiahe Liao[1], Yingbo Yan[1], Wenbin Kang[1], Rongjing Zhang[1], Zemin Liu[1], Tianlu Wang[1] & Metin Sitti[1,8,9 ✉]

Magnetic soft robots offer considerable potential across various scenarios, such as biomedical applications and industrial tasks, because of their shape programmability and reconfigurability, safe interaction and biocompatibility[1–4]. Despite recent advances, magnetic soft robots are still limited by the difficulties in reprogramming their required magnetization profiles in real time on the spot (in situ), which is essential for performing multiple functions or executing diverse tasks[5,6]. Here we introduce a method for real-time in situ magnetization reprogramming that enables the rearrangement and recombination of magnetic units to achieve diverse magnetization profiles. We explore the applications of this method in structures of varying dimensions, from one-dimensional tubes to three-dimensional frameworks, showcasing a diverse and expanded range of configurations and their deformations. This method also demonstrates versatility in diverse scenarios, including navigating around objects without undesired contact, reprogramming cilia arrays, managing multiple instruments cooperatively or independently under the same magnetic field, and manipulating objects of various shapes. These abilities extend the range of applications for magnetic actuation technologies. Furthermore, this method frees magnetic soft robots from the sole reliance on external magnetic fields for shape change, facilitating unprecedented modes and varieties of deformation while simultaneously reducing the need for complex magnetic field generation systems, thereby opening avenues for the development of magnetic actuation technologies.

Soft robots, characterized by their shape programmability, high compliance and physical adaptability, can safely and efficiently interact with complex environments[1,3,7]. They have demonstrated marked potential in various application scenarios, including healthcare[8–11], industry[12,13], marine exploration[14,15] and search and rescue operations[16]. The morphing and functionality of soft robots primarily rely on their actuation components, with common forms of actuation, including light[17,18], heat[19,20], sound[21], magnetic fields[1,22,23], fluids[24,25], electric fields[26–28] and tendon–cable mechanisms[29]. Among these, magnetic actuation is one of the most promising actuation strategies for soft robots in confined operation spaces[1–4]. In magnetic soft robots, deformation arises from the interaction between the external magnetic field and the intrinsic magnetization profile of the robot. It is essential to be able to reprogram the magnetization profile in real-time on the spot (in situ) during their applications requiring diverse tasks or multiple functions[5,6]. Existing studies have explored reprogramming techniques[5,30–36], but they failed to achieve real-time in situ magnetization reprogramming (Supplementary Note 1). We observe that sunflowers in nature achieve adaptive deformation by redistributing internal auxin in response

to changes in solar position (Supplementary Fig. 1). Inspired by this biological phenomenon, we propose a real-time in situ magnetization reprogramming approach that achieves deformation by redistributing internal magnetic units.

## Real-time in situ magnetization reprogramming

Our proposed approach modifies the relative positions of magnetic units within a soft robot by dynamically and locally manipulating their carriers (Supplementary Fig. 1), thereby enabling real-time in situ reprogramming of the magnetization profile of the robot. A magnetic unit is defined as a collection of magnetic material within a continuous region that exhibits uniform magnetization direction. This unit is independent of its volume size and is defined solely by the magnetization. For example, in Fig. 1a(ii), the black region at the end of the tube, which has a downward magnetization direction, constitutes one magnetic unit (diameter, 1.9 mm; length, 2 mm). By contrast, in Fig. 1a(i), the tube shows a helical magnetization profile in which magnetic particles along its axis have varying magnetization directions, with each

[1]Physical Intelligence Department, Max Planck Institute for Intelligent Systems, Stuttgart, Germany. [2]State Key Laboratory of Digital Medical Engineering, School of Instrument Science and Engineering, Southeast University, Nanjing, China. [3]Jiangsu Key Laboratory of Robot Perception and Control, School of Instrument Science and Engineering, Southeast University, Nanjing, China. [4]Institute for Biomedical Engineering, ETH Zurich, Zurich, Switzerland. [5]State Key Laboratory of Fluid Power and Mechatronic Systems, School of Mechanical Engineering, Zhejiang University, Hangzhou, China. [6]Liangzhu Laboratory, School of Mechanical Engineering, Zhejiang University, Hangzhou, China. [7]School of Mechanical Engineering and Hebei Provincial Key Laboratory of Heavy Machinery Fluid Power Transmission and Control, Yanshan University, Qinhuangdao, China. [8]College of Engineering, Koç University, Istanbul, Turkey. [9]School of Medicine, Koç University, Istanbul, Turkey. [10]These authors contributed equally: Xianqiang Bao, Fan Wang, Jianhua Zhang. ✉e-mail: msitti@ku.edu.tr

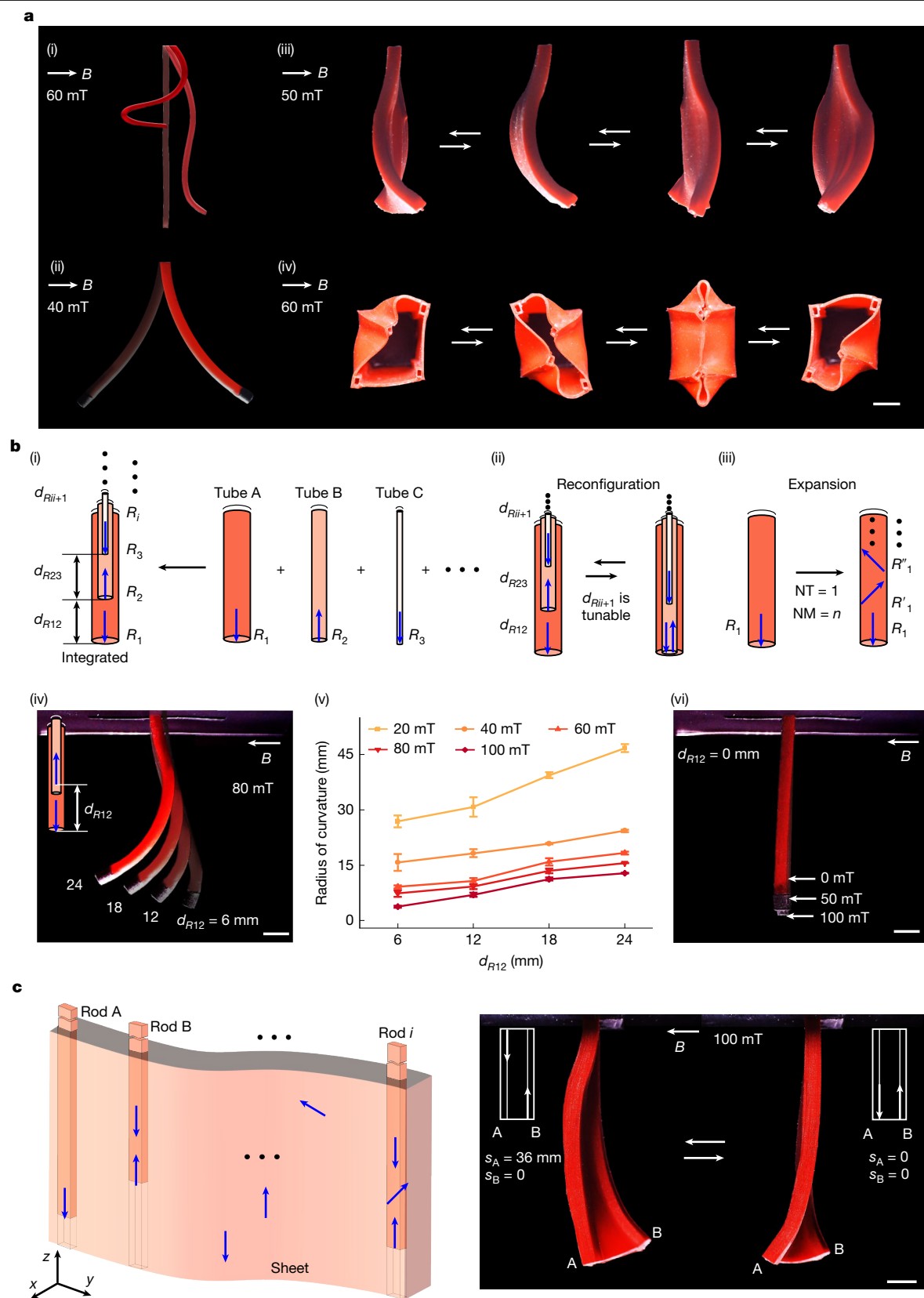

**Fig. 1 | See next page for caption.**

particle representing a separate magnetic unit (diameter of about 0.05 mm). Magnetic unit carriers refer to components in soft robots that are capable of transporting magnetic units. They can take the form of nested tubes (Fig. 1b(i) and Supplementary Fig. 1) or built-in rods (Fig. 1c (left) and Supplementary Fig. 25), and they undergo movement under the influence of external forces. This approach embodies a

**Fig. 1 | Illustration of the real-time in situ magnetization reprogramming method. a**, Under a uniform magnetic field, several demonstrations of deformation are achieved through the proposed reprogramming method, ranging from (i) 1D tubes to (ii) 2D sheets and progressing to (iii) 3D box-shaped structures, with transformations occurring at different moments (Supplementary Videos 1 and 2). All objects are suspended, with (i)–(iii) shown in the front view and (iv) in the bottom view. In (i), the tube transitions from a straight line into a helix without changing the magnetic field. The colour of the tube in (i) was digitally adjusted from black to red for enhanced visualization, whereas the objects in (ii)–(iv), originally red, were left unmodified. **b**, The 1D magnetization reprogramming. (i) Multiple tubes, each embedded with a magnetic unit, undergo a nested assembly to form an integrated tube. $R_i$ represents magnetic unit $i$, in which the blue arrow indicates the magnetization

direction of the magnetic unit. $d_{Rii+1}$ indicates the distance between the magnetic units $i$ and $i + 1$. (ii) The reconfiguration is achieved through the relative movement between tubes, resulting in different magnetization profiles. (iii) Magnetization expansion of a single tube. In (i), each tube contains one magnetic unit, which can be expanded to $n$ magnetic units with varying magnetization directions. NT, number of tubes; NM, number of magnetic units. (iv)–(vi) Shape change of configuration B. Data in (v) are presented as mean ± s.d. ($n = 3$ tests). In (vi), when $d_{R12} = 0$ mm, the tube undergoes negligible deformation as the magnetic field increases from 0 mT to 100 mT. **c**, The 2D magnetization reprogramming. Left, the number and positions of the magnetic units within the sheet can be arbitrarily configured and altered. Right, shape changes of sheets with two different magnetic unit positions (Supplementary Video 2). Scale bars, 5 mm.

universal framework that can be adapted to various configurations of soft robots and their components. It accommodates diverse forms of soft robotic systems, such as tubular robots (Fig. 1b) or sheet-shaped robots (Fig. 1c), among others. The magnetic units integrated within this framework may vary widely, from magnetic particles (Fig. 1a(i)) to permanent magnets (Fig. 1a(iii)), each differing in size and magnetization capacities. This approach allows for the customization or modification of the positions of magnetic units according to various requirements, thereby generating different magnetization profiles to achieve diverse deformations. For instance, as shown in Fig. 1a, various shapes, such as helical and linear shapes under the same uniform magnetic field, along with their real-time interconversion, are shown. These deformation modes are unachievable through existing methodologies. The method of reconfiguring magnetic unit positions can be applied to configure a singular magnetic unit (Fig. 1a(ii)) or multiple magnetic units (Fig. 1a(iii),(iv)), and even an infinite number of magnetic units (Fig. 1a(i)). It can be used in magnetization reprogramming in one-dimensional (1D) (Fig. 1a(i),(ii)), two-dimensional (2D) (Fig. 1a(iii)) and three-dimensional (3D) (Fig. 1a(iv)) robot morphologies. Moreover, when combined with tunable magnetic fields, they can broaden the range and diversity of shape deformations (Supplementary Fig. 71 and Supplementary Note 8).

For the 1D magnetization reprogramming, we selected a tubular robot, a widely used representative, as the platform, as shown in Fig. 1b(i). We used a nested approach with multiple tubes, each containing preprogrammed magnetic units. The relative positions between tubes ($d_{Rii+1}$) are tunable according to requirements (Fig. 1b(ii)). By altering $d_{Rii+1}$, we facilitated changes in the positions of the magnetic units, thereby achieving various magnetization profiles. Moreover, the number of magnetic units in each tube can be expanded as needed (Fig. 1b(iii)). We explored all possible configurations (configurations A–G) and shapes of the tubular robot (Supplementary Figs. 2–15), demonstrating the possibility of generating multiple deformations in real-time and in situ by using this proposed method without the need for altering the magnetic field. Each configuration exhibits distinct characteristics and abilities. For instance, configuration A (two tubes, inner tube with one magnetic unit) enables selective activation of magnetism in the robot (Supplementary Figs. 3 and 4, with specific applications in section 'Reprogrammable cilia array'); configuration C (two tubes, inner tube with multiple magnetic units) possesses shape retention abilities (Supplementary Figs. 9–11, with specific applications in the next section); configuration G (two tubes, each tube with countless magnetic units) enables rapid switching between various complex deformations, such as linear and helical shapes (Supplementary Fig. 15 and Supplementary Video 1, also with specific applications in the next section). Under the same configuration, different shapes were also achieved by altering the relative positions of the tubes. For instance, in configuration B (two tubes, each tube with one magnetic unit), within a uniform magnetic field of 80 mT, as the displacement of the magnetic units ($d_{R12}$) varies from 0 mm in increments of 6 mm up to 24 mm, the radius of curvature of the tube changes from +∞ to 7.3 mm

and then to 15.6 mm, whereas the terminal bending angle shifts from 0° to 21.5° and then to 64.7° (Fig. 1b(iv),(v) and Supplementary Fig. 6). The radius of curvature and terminal bending angle are specifically defined to describe the shape changes of the tube (Supplementary Fig. 29). Moreover, within various magnetic fields, more shape changes were obtained under the same configuration. For example, in configuration B with $d_{R12}$ set at 6 mm, the increase in magnetic field intensity from 20 mT to 100 mT leads to a decrease in the radius of curvature from 26.9 mm to 3.8 mm and an increase in the terminal bending angle from 3.9° to 26.3° (Supplementary Fig. 7).

For the 2D magnetization reprogramming, we selected a sheet-shaped robot as the platform. This robot is designed with strategically placed holes to facilitate the movement of rods, which are equipped with preprogrammed magnetic units within its structure (Fig. 1c). We developed four distinct configurations (configurations A–D) and evaluated their abilities to undergo shape transformations (Fig. 1a(iii), Supplementary Figs. 16–24 and Supplementary Video 2). Similarly, we extended our proposed method to 3D magnetization reprogramming. We created a soft 3D box-shaped robot, in which different deformations are achieved in real-time under the same magnetic field by adjusting the layout of magnetic units (Fig. 1a(iv), Supplementary Figs. 25 and 26 and Supplementary Video 2). The 2D and 3D magnetization reprogramming are spatial expansions of 1D magnetization reprogramming. Considering the wide array of deformations and the characteristics already demonstrated in 1D magnetization reprogramming, the 2D and 3D reprogramming methods enable a spatial expansion of these deformation characteristics, thus facilitating the emergence of diverse properties.

During our experiments, we found two distinct characteristics: (1) magnetic neutralization and (2) magnetic reversal. Magnetic neutralization happens when two magnetic units with identical magnetization magnitudes but opposite directions (that is, the magnetic moments **m** and −**m**) are axially aligned within the tube, their magnetizations cancel each other out overall. In this configuration, the tube appears non-magnetized externally, and the magnetic forces are effectively neutralized, thereby preventing any deformation. For example, in Fig. 1b(vi) and Supplementary Fig. 8a, the tube remains upright with almost no change in shape as the magnetic field increases (minor changes being attributable to manufacturing errors in the magnetic units). Using this characteristic, we managed to achieve a substantial transformation of the tube from a straight to a helical shape without altering the magnetic field (Fig. 1a(i), Supplementary Fig. 15 and Supplementary Video 1). Furthermore, to eliminate the influence of changes in stiffness due to the movement of internal rods, we carried out comparative experiments that verified these changes did not affect the deformations (Supplementary Fig. 15e,f and Supplementary Video 1). This feature allows for the magnetization of soft robots to be selectively activated or deactivated, thereby eliminating or initiating the influence of the external magnetic field on magnetic robots. This control makes it possible to selectively manipulate a specific magnetic robot within a field that includes multiple magnetic robots. Potential applications include coordinated multi-instrument operation within the same magnetic field

(see section 'Coordinated multi-instrument operation'). Moreover, relocating the two magnetic units from an overlapping configuration to distinct positions regenerates the magnetization profile of the tube, which, in turn, produces various shapes. Magnetic reversal occurs when a −2**m** magnetic unit aligns axially with an **m** magnetic unit inside the tube: the magnetization strength of the target unit remains unchanged, but its direction of magnetization is reversed (that is, changing from **m** to −**m**). As shown in Fig. 1a(ii) and Supplementary Fig. 8b, without altering the magnetic field, the bending direction of the tube switches from left to right. In the deformations shown in Fig. 1a(ii), the degrees of bending to the left and right are not identical, which is due to changes in stiffness caused by the introduction of an inner tube. The magnetic moment of the inner tube can be set to −2**m** + Δ**m**, where Δ**m** compensates for the stiffness changes resulting from the introduction of the inner tube. This ability to completely reverse the deformation direction of the tube without altering the magnetic field offers new possibilities for complex operations within the same magnetic field (with specific application in the section 'Reprogrammable cilia array').

In practical applications, these configurations can be used alone or in combination. The force used to actuate relative motion among magnetic unit carriers can originate from a wide range of actuators—such as motors (Supplementary Fig. 87); pneumatic, hydraulic or piezoelectric devices; or from any form of motion-generating stimulus. We provide the theoretical analysis, simulation calculations, manufacturing details, various potential analyses, magnetization profile design method and specific design schemes and tuning strategies for various application scenarios in the Methods, Supplementary Notes 2–11 and Supplementary Figs. 2–96. In the following subsections, we examine four applications to demonstrate the potential of the proposed method, which extends beyond these examples.

## Contact-free object navigation

Minimally invasive surgery is now widely used because of its benefits in reducing tissue trauma, decreasing postoperative recovery time and lowering the risk of complications[37–40]. In various types of minimally invasive surgery, minimally invasive instruments inevitably collide with or contact tissues (Fig. 2a(i),(ii)), thereby giving rise to multiple potential risks (discussed in Supplementary Note 1). To address the limitations of existing research, we proposed a method based on the principles demonstrated by configuration C in 1D magnetization reprogramming. For ease of manufacture, we used rods to replace the inner tubes, and by nesting these inner rods within the outer tubes, we assembled the required tubes (catheters). To achieve different deformations, two types of tube configurations were used. The first configuration features a rod with two magnetic units, whereas the second configuration includes a rod with three magnetic units (Supplementary Fig. 31a,b).

In Fig. 2b(i), under a uniform magnetic field, the soft tube in the control setup (existing method, Supplementary Fig. 31c) was pushed forward or backwards, and its bend moved with it, preventing it from bypassing avoidance objects without contact. By contrast, with our proposed method (first configuration, Supplementary Fig. 31a), the bend in the soft tube remained stationary regardless of whether the soft tube moved forward or backwards (Fig. 2b(ii)). This feature of the proposed method facilitates contactless navigation over avoidance objects. Moreover, Fig. 2b(iii) shows a soft tube (second configuration, Supplementary Fig. 31b) that forms two bent sections, enabling it to bypass more intricate avoidance objects without contact. The aforementioned operations are demonstrated in Supplementary Video 3.

To further validate the feasibility of the proposed method, we condensed the scenario in Fig. 2a(i) into Fig. 2a(iii). In this scenario, the soft tube must navigate a designated path around the avoidance object without contact. During this process, the magnetization profile of the soft tube needs to be modified in real-time. We divided the soft tube into multiple small segments, labelled as S1, S2, … S$i$. Each segment

state is classified as either 0 or 1, where 0 indicates a non-magnetic state and 1 indicates a magnetic state (Supplementary Fig. 32). Supplementary Fig. 33 shows the state changes in the four segments of the soft tube necessary to complete the task. Using the existing method, it is challenging to achieve real-time modifications of the magnetization profile, thereby hindering the completion of the task above. Although theoretically possible through the superposition of multiple magnetic fields, this approach is overly complex and primarily of theoretical interest, with limited practical applicability and no documented cases in existing literature. To illustrate the above operations, we designed an experimental setup (Supplementary Fig. 34) based on Fig. 2a(iii). Using the existing methods, contact with the avoidance objects was not possible (Supplementary Video 4). However, by using the proposed method, we easily achieved contact-free navigation around the avoidance object to reach the target location (Fig. 2c). We defined the distances from the two magnetic units in the soft tube to the end of the soft tube as $s_A$ and $s_B$ to describe the changes in the magnetization profile during the process (Fig. 2d(i)). We observed that the position of the magnetization profile changes in real-time with the movement of the soft tube (the light-blue area in Fig. 2d(ii), approximately 20–32 s), but the relative position of the two magnetic units ($s_A − s_B$) remains unchanged, thereby enabling the possibility of navigating around the avoidance object. Furthermore, we extended the above experiments by setting up more complex scenarios (Supplementary Fig. 35), and the task was also completed (Fig. 2e). The magnetic fields and demonstrations of the two operations are presented in Supplementary Fig. 36 and Supplementary Video 4, respectively.

We also designed a vascular model with three branches having different curvature radii and bending angles (branches A, B and C) (Supplementary Fig. 37). By applying a magnetic field in the same direction and gradually changing the magnitude (Supplementary Fig. 38), the soft tube was able to enter and exit the three branches without any contact (Fig. 2f, Supplementary Fig. 39 and Supplementary Video 5)—achieving the ideal contact scenario discussed in Supplementary Note 5 and Supplementary Fig. 30c, in which there is contact only with the bottom and not with the sides. For comparison, we also demonstrated the performance of existing methods in the vascular model, which inevitably resulted in contact (Supplementary Video 5). Furthermore, we conducted ex vivo animal experiments using a porcine heart (Supplementary Figs. 40 and 41 and Supplementary Video 6). We found that the ideal contact scenario was successfully achieved using the proposed method.

To validate the ability of the proposed method for achieving 3D contact-free object navigation, we fabricated two distinct 3D experimental scenarios (Supplementary Figs. 81 and 82). By designing different magnetization profiles, we achieved contact-free navigation through these 3D objects under a single magnetic field (Fig. 2h, Supplementary Figs. 80 and 83 and Supplementary Video 17). In future practical applications, there will probably be increased demands on the operational length, the number of bends, the curvature and the number of layers. To address these specific challenges, we proposed four new methods—a segmented design method, a multi-magnetic field control method, a multi-bend deformation design method and a zone-based tube nesting design method—targeting long-range operation, multi-bend deformation, high-curvature deformation and multi-nested tube operation, respectively. For example, regarding the long-range operation, we demonstrated that a tube with a length of 305 mm could be manipulated while achieving multiple bending deformations (Supplementary Figs. 84 and 85 and Supplementary Video 18). This length can be further increased with enhanced stiffness of tubes. We integrated commercial catheters for manipulation, demonstrating an achievable operational length of at least 1,240 mm (Supplementary Fig. 86 and Supplementary Video 19). We believe that the aforementioned operational distances essentially satisfy most application requirements. The detailed designs, tuning strategies

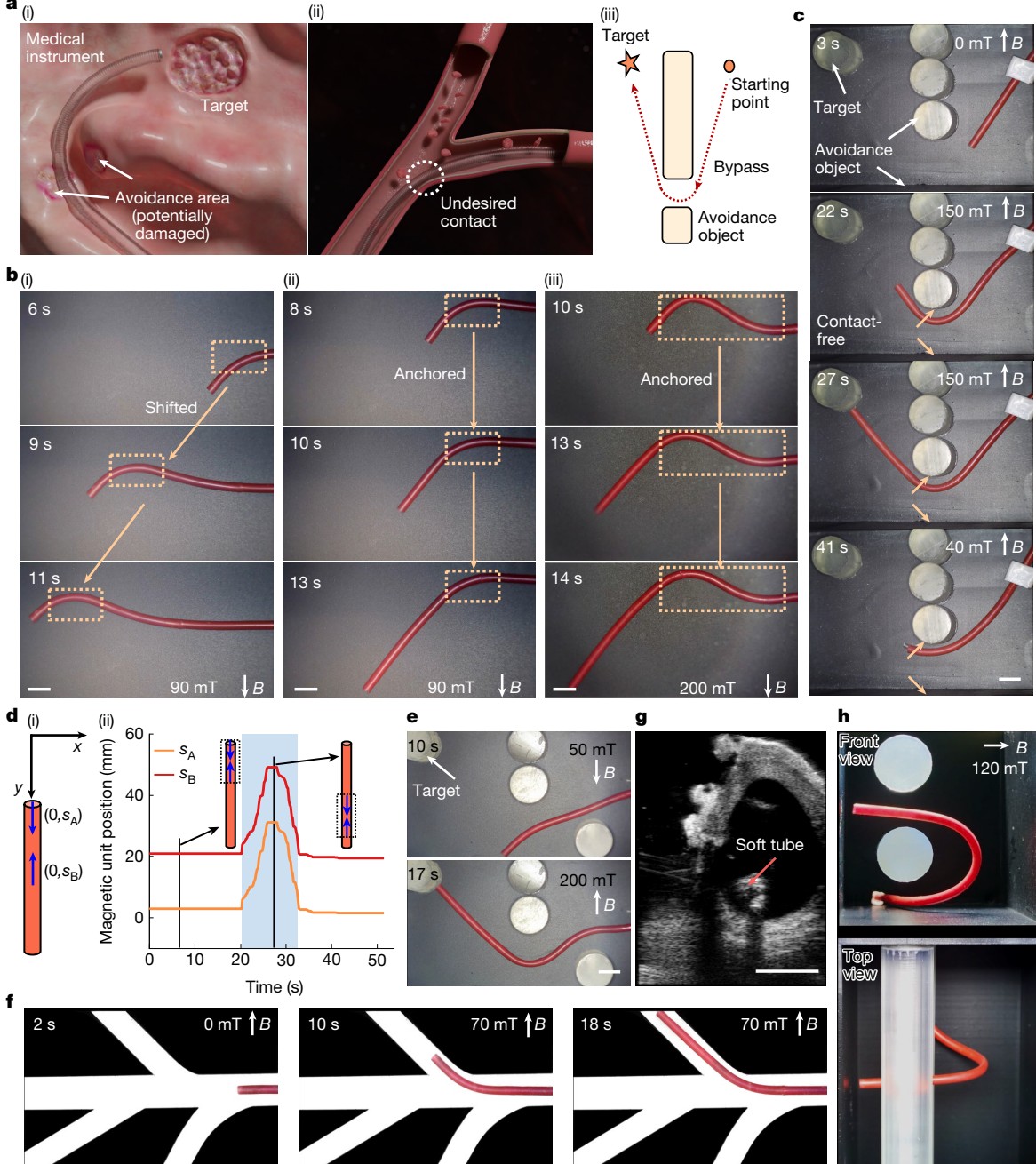

**Fig. 2 | Demonstration of contact-free object navigation. a**, Two typical medical scenarios. (i) A medical instrument traverses through avoidance tissues to reach the target area for treatment. (ii) A catheter navigates through blood vessels, unable to avoid contact. (iii) An abstract representation of bypassing the avoidance object in (i). **b**, Video snapshots illustrating the principles of three different operating modes (Supplementary Video 3). (i) Using the existing method, the position of the bend in the soft tube changes as it moves within a uniform magnetic field. (ii),(iii) The proposed operating modes, in which the soft tube forms one or two bends in a uniform magnetic field, with the position of the bends remaining anchored during the movement. **c**, Video snapshots showing the soft tube reaching and retracting from the

target position without contact, corresponding to the scenario in **a**(iii) (Supplementary Video 4). **d**, Partial parameters corresponding to the operation process in **c**. (i) The magnetization profile of the soft tube quantified using $s_A$ and $s_B$. (ii) Changes in the magnetization profile during the operation process. **e**, Video snapshots of the soft tube deforming with two bends to bypass avoidance objects (Supplementary Video 4). **f**, Video snapshots showing the soft tube navigating through blood vessel models without contact (Supplementary Video 5). **g**, Ultrasound imaging of a soft tube navigating through the vasculature of an ex vivo porcine heart (Supplementary Video 6). **h**, The 3D dual-obstacle traversal under a single magnetic field (Supplementary Video 17). Scale bars, 5 mm.

and related discussions of these methods are presented in section 4 of Supplementary Note 9. This approach, which avoids contact with the surrounding environment, could be applied to or inspire other medical diagnostic and treatment methods, as well as industrial applications.

## Reprogrammable cilia array

The ubiquitous cilia in nature play an important part in various creatures, including human beings, because of their essential functionalities, such as cerebrospinal fluid transportation[41], mucus clearance[42],

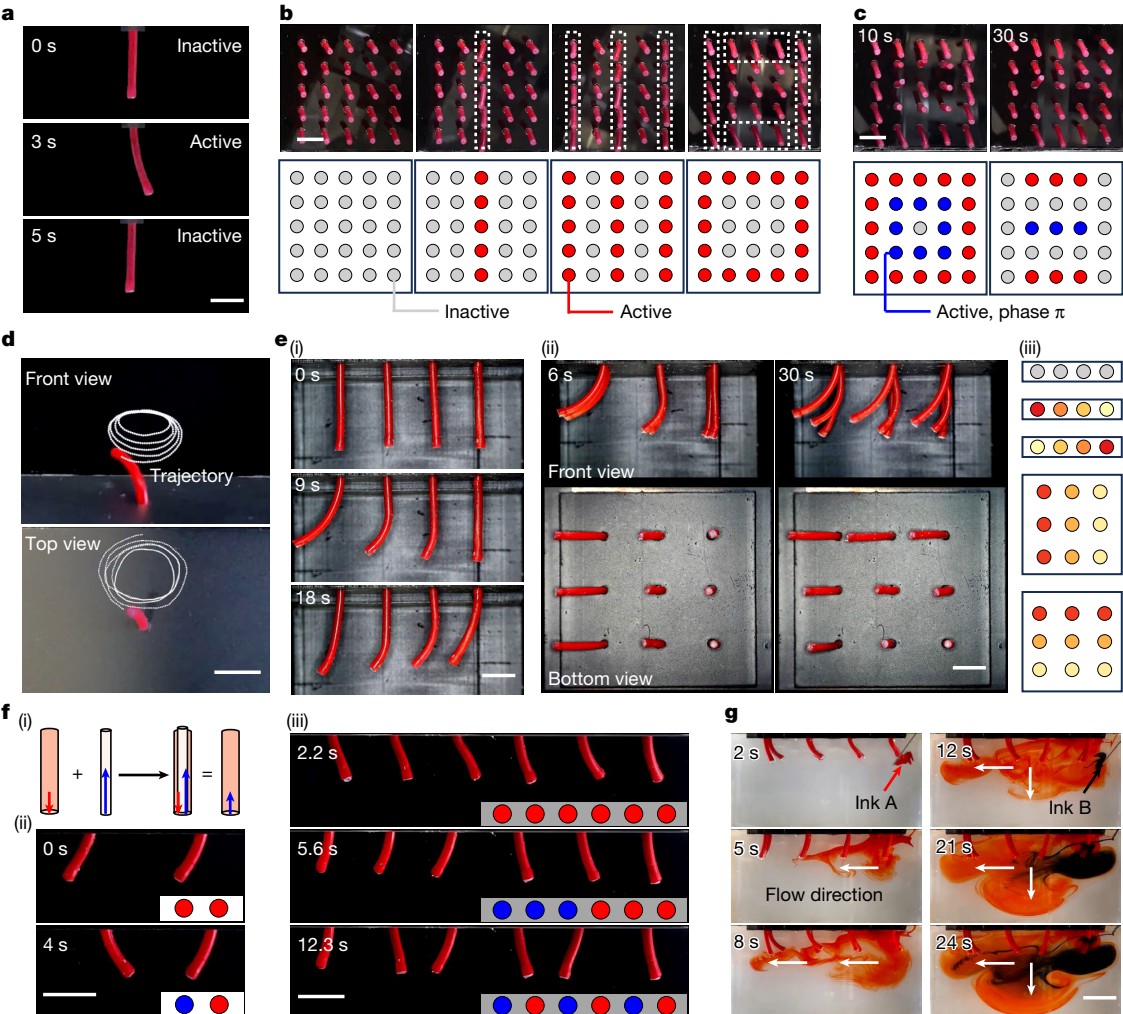

**Fig. 3 | Reprogrammable cilia array demonstration. a**, Display of the activation of a single cilium. **b**, Targeted cilia activation. The photos in the top row are video snapshots of the operation (Supplementary Video 7), and diagrams in the bottom row show the state of cilia corresponding to the photos above. The grey and red circles indicate inactive and active cilia, respectively. **c**, Video snapshots of cilia states following the integration of the targeted cilia activation with the ciliary phase modulation (Supplementary Video 7). The symbols in these figures are similar to those in **b**, in which the blue circles represent cilia activated with a phase difference of π relative to the red active cilia. **d**, Ciliary bending amplitude modulation under the same rotating magnetic field (Supplementary Video 8). The dashed lines show the tip trajectory of the cilium. **e**, Ciliary bending amplitude modulation under a uniform magnetic field (Supplementary Video 8): (i) 1 × 4 array; (ii) 3 × 3 array; and (iii) diagrams showing the ciliary states corresponding to the snapshots in (i) and (ii), with the first three representing the states from (i) and the latter two representing states from (ii). The intensity of the circle colour indicates the bending amplitude magnitude, with red denoting the highest amplitude and grey indicating a ciliary amplitude of zero. **f**, Ciliary phase modulation: (i) the underlying principle of ciliary phase modulation; (ii) video snapshots showing phase changes in two cilia; and (iii) video snapshots showing phase changes in six cilia (Supplementary Video 9). The schematics in the bottom right corner indicate the phase of the cilia, with red circles representing cilia at phase 0 and blue circles denoting cilia at phase π. **g**, Video snapshots demonstrating the application of ciliary phase modulation, which can redirect fluid flow from leftward to both leftward and downward directions (Supplementary Video 9). Scale bars, 5 mm.

egg cell delivery[43], self-propulsion[44], food capture[45] and self-cleaning[46]. The inability to achieve in situ reprogrammability constrains the diverse functionality of cilia (Supplementary Note 1). To address this issue, we proposed a method for reprogramming cilia based on configurations A and B of the 1D magnetization reprogramming. This method exhibits the possibility of reprogramming cilia in three ways: targeted cilia activation, ciliary bending amplitude modulation and ciliary phase modulation.

In the targeted cilia activation, we used configuration A from the 1D magnetization reprogramming. By using the properties at $d_{R12} = 0$ mm and $d_{R12} = 42$ mm, we were able to selectively magnetize the cilia (Supplementary Figs. 3c and 4d). To facilitate manipulation and observation, we fabricated millimetre-scale cilia (Supplementary Fig. 42). Figure 3a shows the ciliary activation in a single cilium: the state of the cilium can be switched between active and inactive by controlling the presence of

the magnetic units ($d_{R12}$), enabling transitions between stationary and moving states under the same magnetic field (generated by a Halbach array, 20 mT). Control over individual cilium possesses the ability to be extended to cilia arrays, allowing for selective activation of specific cilia to be active or inactive, as shown in Fig. 3b, Supplementary Fig. 43 and Supplementary Video 7. To facilitate observation, schematics are drawn below the video snapshots. These diagrams show the variety of patterns that can be formed. By using the targeted cilia activation, various states and driving modes of cilia arrays are achieved, offering new possibilities for different functionalities. A potential application of this technique is to transport matters along a desired pathway. It is possible to transport fluids or solids along predetermined trajectories by sequentially activating cilia in specific rows and columns, such as the U-shaped pathway shown in Supplementary Fig. 44 and Supplementary Video 7. This U-shaped pathway demonstrated the

potential of the proposed method to transport matters along arbitrary pathways. However, owing to the preliminary nature of the current study, the specific directions of fluid or solid propulsion remain to be determined, and further in-depth research is required to clarify effective driving directions.

In the ciliary bending amplitude modulation, we used configuration B from the 1D magnetization reprogramming. By varying $d_{R12}$, we achieved various bending amplitudes in a soft tube. This approach allows a single cilium to perform diverse motion patterns in the same rotating magnetic field. To demonstrate this functionality, we fabricated cilia similar to those used in the targeted cilia activation (Supplementary Fig. 45). Figure 3d shows the various motion trajectories of a single cilium actuated by the same rotating magnetic field (Supplementary Fig. 46 and Supplementary Video 8). By altering the parameter $d_{R12}$, we could adjust the bending amplitude of the cilia, achieving five distinct trajectories that enabled different ciliary motion patterns. To visually illustrate these changes in bending amplitude, we demonstrated them in a uniform magnetic field (40 mT) instead of a rotating one. Figure 3e(i),(ii) show the amplitude modulation in a 1 × 4 array and a 3 × 3 array, respectively. To better illustrate the variation in bending amplitude, we used circles with varying shades to represent the amplitude of the cilia (Fig. 3e(iii)), with further details shown in Supplementary Figs. 47 and 48 and Supplementary Video 8. These experiments demonstrated that without altering the external magnetic field, it is possible to generate and switch between different motion patterns for a single cilium and a cilia array.

In the ciliary phase modulation, we used the magnetic reversal characteristic from configuration B in the 1D magnetization reprogramming to achieve alteration in the magnetization direction, thus facilitating the modulation of the cilia phase. As shown in Fig. 3f(i), superimposing a magnetic unit of $-2\mathbf{m}$ onto the original cilium ($\mathbf{m}$) reverses its magnetization to $-\mathbf{m}$. To demonstrate the feasibility of the proposed design, we fabricated cilia prototypes, with design details shown in Supplementary Fig. 49. Figure 3f(ii) shows how two cilia with the same phase under the same rotating magnetic field (45 mT) are altered to have opposite phases. Furthermore, we showcased the phase modulation changes in a 1 × 6 cilia array (Fig. 3f(iii) and Supplementary Fig. 50). These operations demonstrated the feasibility of using the proposed method for phase modulation of cilia arrays, providing a new approach for implementing complex ciliary motion patterns and potentially expanding the applications of cilia. A potential application is the real-time modulation of fluid driving directions. Continuing with the cilia used in previous experiments, we constructed a 2 × 4 cilia array and submerged it in 85% glycerine, driven by a rotating magnetic field of 45 mT. When all cilia were in the same phase, they drove the fluid to move leftward (Supplementary Fig. 51); when the cilia in the left two columns were out of phase by π with those in the right two columns, they were able to drive the fluid to move both leftward and downward (Supplementary Fig. 52). To illustrate the specific process of phase modulation, we initially drove the fluid to move left and then changed the ciliary phase to drive the fluid downward and leftward (Fig. 3g). All the above operations are presented in Supplementary Video 9. In the experiment, square-shaped paddles were attached to the tips of each cilium to enhance the visibility of fluid flow (Supplementary Fig. 53). This design merely amplified the driving effect without changing the driving mode and thus did not affect the experimental outcomes. The ability to control the ciliary phase to change driving directions allows fluid manipulation to be adjusted on demand, enhancing driving efficiency and providing greater operational flexibility. Although the phase difference in the aforementioned cilia is π, we can use more complex designs of magnetization directions to achieve various desired phase differences.

The targeted cilia activation, ciliary bending amplitude modulation and ciliary phase modulation can be combined to enable operations and functionalities that are not possible when these techniques are applied independently. For example, by integrating the targeted cilia activation with the ciliary phase modulation, it is possible to achieve more complex ciliary driving patterns (Fig. 3c, Supplementary Fig. 54 and Supplementary Video 7). The above demonstrations are based solely on configurations A and B from 1D magnetization reprogramming. Other configurations can also be introduced to enable even more complex motion patterns for individual cilia, thus offering greater possibilities for expanding ciliary functions and more complex applications. Moreover, the dimensions of the cilia can be adjusted according to specific operational requirements.

## Coordinated multi-instrument operation

In medical diagnosis and treatment, multiple medical instruments are commonly needed to work together to achieve the desired outcome (Fig. 4a). In magnetic actuation, owing to the pervasive nature of magnetic fields, which affect magnetic objects within their range non-selectively, independently controlling multiple targets using the same magnetic field poses a considerable challenge[47]. Therefore, magnetic actuation methods are often limited to manipulating a single instrument, which restricts their use in scenarios requiring multiple instruments (Supplementary Note 1). The approach proposed in this research provides an effective means to address this challenge. We used the magnetic neutralization feature in configuration B of the 1D magnetization reprogramming to enable the instruments to selectively respond to external magnetic fields as needed. Moreover, by adjusting $d_{R12}$, the instruments can achieve real-time shape transformation, thereby attaining the necessary deformations to accomplish various tasks.

To demonstrate the feasibility of this method, we constructed two types of integrated soft tubes, referred to as tube A and tube B (Supplementary Fig. 55). The inner tube was substituted with a soft rod to facilitate manufacturing and manipulation. Tubes A and B were operated under a uniform magnetic field (Fig. 4b and Supplementary Video 10). We measured the variations in the radius of curvature and the terminal bending angles of the tubes (definition in Supplementary Fig. 29) throughout the operation (Fig. 4c). Note that, in their straight state, the radius of curvature is infinite, and the terminal bending angle is 0°. We observed that the two tubes can each undergo arbitrary deformations at any given moment without mutual interference. The magnetization profile settings for the two tubes are relatively simple, resulting in straightforward deformations. More complex deformations or functionalities can be achieved by using more complex magnetization designs.

To further explore the potential application of this proposed method, we established three experimental scenarios based on typical application settings shown in Fig. 4a. The instruments used in these operations are the same as those in Fig. 4b. The first experimental scenario (Supplementary Fig. 56a) involves numerous tissues between the target and the entry site, which act as obstacles to diagnostic or therapeutic procedures. The target requires the coordinated action of two instruments: one instrument follows path A to the target to complete preliminary treatment tasks; subsequently, the other instrument follows path B to the target to administer adjunctive therapy. This process is shown in Supplementary Video 11. Throughout this process, both tubes continuously adapt their shapes in real-time to execute different operations. In the second experimental scenario, two targets were established (Supplementary Fig. 56b); these targets require sequential operations and treatments, necessitating three instruments, which follow paths A, B and C to reach targets A and B. Details of the specific operations can be viewed in Supplementary Video 12. In the third experimental scenario (Fig. 4d), a target was set up, requiring one instrument to perform auxiliary operations near position A along path A, and the other instrument to proceed along path B to carry out diagnostic and therapeutic operations at the target location. However, path B is

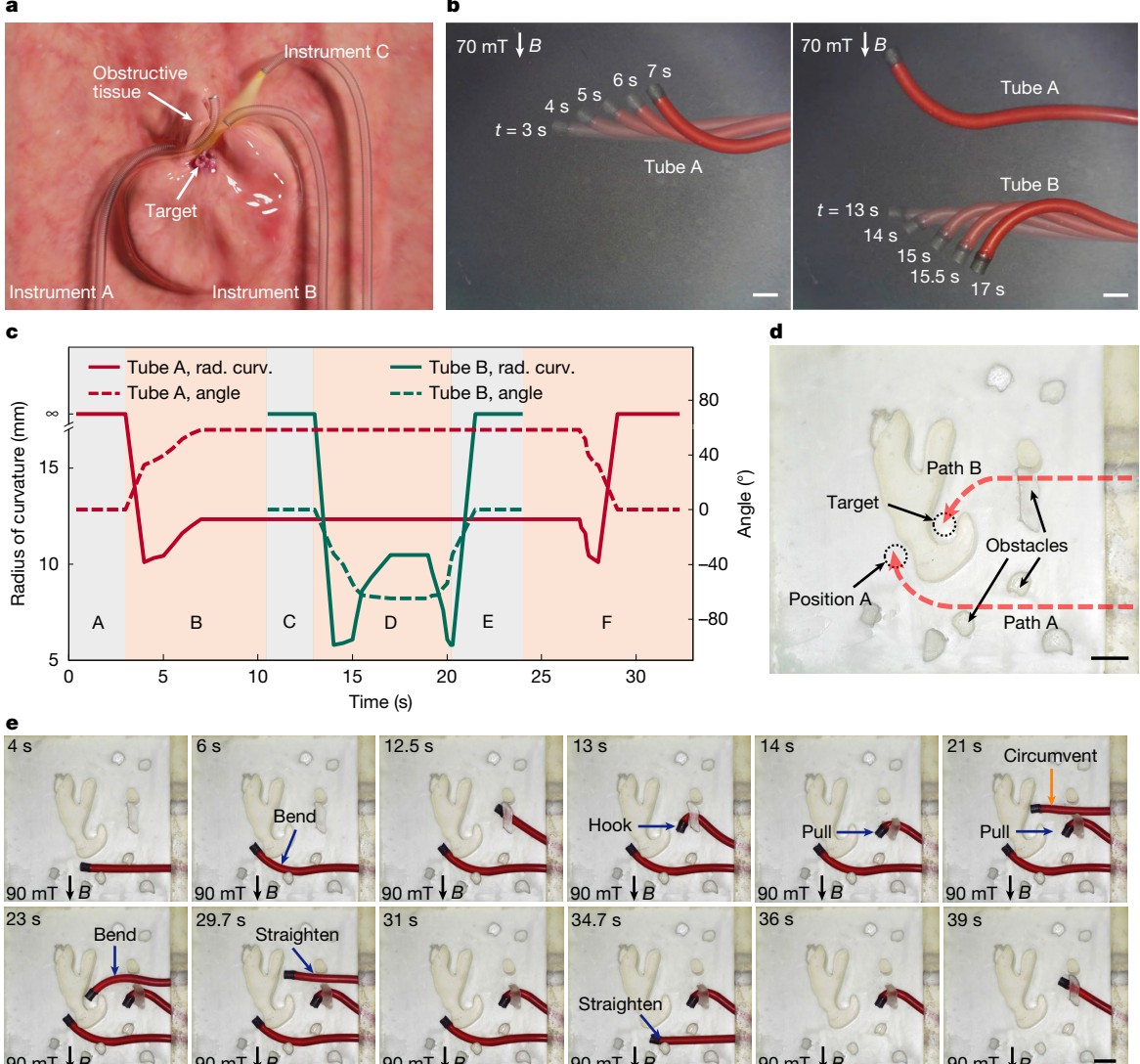

**Fig. 4 | Coordinated multi-instrument operation demonstration.**
**a**, A typical multi-instrument-coordinated operational medical scenario, in which the diagnostic or treatment target is obscured by tissue and multiple instruments need to be operated simultaneously. **b**, Independent control of tubes A and B under the same uniform magnetic field, illustrated through video snapshots (Supplementary Video 10): under a uniform magnetic field of 70 mT directed downward (left), tube A achieves various shapes; in the same uniform magnetic field, tube A maintains a specific shape whereas tube B achieves different shapes as required (right). **c**, Description of the shapes of tubes A and B during the operation process in **b**. This figure is segmented into six zones (A–F) using grey and orange. The radius of curvature (rad. curv.) and the terminal bending angle are used to describe the shapes (Supplementary Fig. 29). **d**, Task setup illustration for the third application scenario. **e**, Video snapshots of the operation in the third application scenario (Supplementary Video 13). Scale bars, 5 mm.

obstructed by barriers, necessitating an additional instrument to clear the pathway. Figure 4e and Supplementary Video 13 demonstrate the entire procedure. For further operational details and discussion, refer to Supplementary Note 12.

In the above operations, because the magnetic field direction remains constant, the resulting operation is confined to a single plane. In practical applications, however, it may be necessary for the tube to operate beyond this plane (that is, 3D operation). One approach is to rotate the external magnetic field; however, this method is effective only for specific tasks, as changes in the magnetic field direction may interfere with the functioning of other tubes (medical instruments). A more effective solution is to adopt configuration G of the 1D magnetization reprogramming. This configuration enables 3D deformation under a uniaxial constant magnetic field (Supplementary Video 16). By combining this configuration with the magnetic neutralization feature, we can achieve both diverse 3D deformation modes and non-deformation states under

a constant magnetic field. Beyond medical applications, this method also holds the potential for other scenarios, such as micro-assembly.

## Shape-adaptable soft grippers

Compared with traditional grippers, soft grippers exhibit unique advantages in interacting with objects because of their excellent compliance and adaptability. For magnetic soft grippers, achieving specific shapes through external magnetic fields results in a limited number of shapes, satisfying only highly specialized needs. The optimal approach involves modifying the magnetization profile of the soft gripper itself to achieve the necessary deformations for effective gripping—this method is termed as real-time in situ magnetization reprogramming, as proposed in this study. For a detailed analysis, see Supplementary Note 1.

Based on the principles of configuration B (sheet without magnetic units; two adjacent rows of rods, each with one magnetic unit) in the

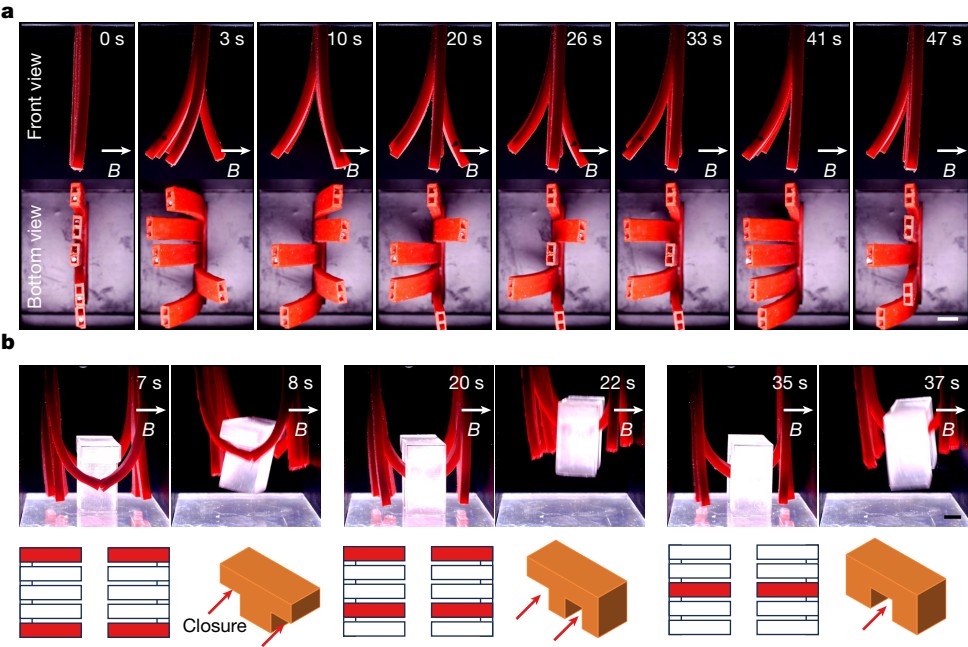

**Fig. 5 | Shape-adaptable gripping demonstration. a**, The hand-like soft gripper achieves various bending modes under the uniform magnetic field of 35 mT directed to the right, as shown in video snapshots of the operation (Supplementary Video 14). **b**, Two hand-like grippers are placed side by side, accomplishing the grasping of different objects. The three sub-figures each show the grasping of three distinct object shapes. All operations are conducted under a uniform magnetic field, with the direction pointing right and a magnitude of 60 mT. Each sub-figure includes two video snapshots (Supplementary Video 15), a schematic describing the state of the gripper and another showing the shape of the object being grasped. In the schematic showing the state of the gripper, the two sections on the left and right correspond to the two hand-like grippers. Each section consists of five rectangles representing the five fingers of the hand-like gripper. The white portions indicate fingers in a relaxed position, and the red portions denote fingers in a bent position (grasping state). Left, grasping a T-shaped object; middle, grasping an F-shaped object; right, grasping an n-shaped object. Scale bars, 5 mm.

2D magnetization reprogramming, we developed a shape-adaptable soft gripper. The gripper is hand-like, featuring five strip-shaped fingers, each of which can bend forwards or backwards independently or in combination under the same magnetic field (Fig. 5a). We verified the ability of the gripper to flexibly bend like a human hand under a uniform magnetic field (Fig. 5a and Supplementary Video 14). The gripper achieved various bends by altering the positions of magnetic units within each finger, which changes its magnetization profile in the same magnetic field. To verify the ability of the proposed gripper to handle objects of various shapes, we simulated human hand functionality by positioning two grippers side by side, allowing them to collaborate in performing diverse gripping tasks (Fig. 5b). Furthermore, we fabricated three objects with different shapes: a T-shaped object, an F-shaped object, and an n-shaped object. These objects were grasped under a uniform magnetic field to demonstrate the feasibility of gripping these varied forms (Fig. 5b and Supplementary Video 15). To grasp the T-shaped object, the two hand-like grippers extended their first and fifth fingers simultaneously, matching the shape of the T-shaped object, thus completing the grip. Similarly, when gripping the F-shaped and n-shaped objects, the two hand-like grippers extended the corresponding fingers to complete the grips. These operations demonstrated that the shape-adaptable soft grippers can effectively grasp objects of different shapes under the same magnetic field.

## Discussion

As we broaden the scope of this research, the following considerations will be valuable for advancing the application of our proposed technologies. First, it is necessary, where needed, to consider the impact of stiffness variations caused by positional changes between the components of the soft robot on its deformation. For instance, when using the magnetic reversal feature in configuration B of the 1D magnetization reprogramming, the degrees of bending to the left and right are not identical (Fig. 1a(ii)). Although this issue has not been extensively discussed in applications, we have provided modelling methods and solutions. These are optional and become necessary only when precise deformation control is considered. Second, the form and size of the magnetic units should be considered in relation to the demands of specific application scenarios, striking an appropriate trade-off. The form and size of the magnetic units directly affect the precision of the magnetization profile generation. Smaller dimensions lead to higher precision in creating these profiles; however, the complexity of fabrication and control increases accordingly. For instance, in configuration G of the 1D magnetization reprogramming, the magnetic units consist of magnetic particles approximately 50 μm in size. In section 'Reprogrammable cilia array', the magnetic units are permanent magnets sized with a diameter of 0.75 mm × 1 mm. Although the precision of magnetization profiles generated using permanent magnets is lower than that achieved with 50-μm magnetic particles, both configurations satisfy the application requirements adequately. Third, complex deformations may require a greater number of magnetic unit carriers, which, in turn, necessitates additional actuators for control. An increase in actuator number compounds the complexity of the actuation and control systems and raises costs. An effective strategy to reduce the number of actuators is to use configurations with more sophisticated, preprogrammed magnetization profiles (for example, configuration G in 1D magnetization reprogramming). We provide a detailed methodology for designing magnetization profiles in Supplementary Note 10. Although fewer actuators simplify the control architecture, complex preprogrammed magnetization profiles increase both design and fabrication challenges, necessitating a careful balance between profile complexity and actuator quantity. Finally, possible solutions may need

to be considered when applying this method on a smaller scale. For miniaturization, the proposed method easily applies at the millimetre and sub-millimetre scale (Supplementary Video 20). However, further size reduction introduces three main issues: difficulties in fabricating soft robotic components, increased friction between components and reduced positional accuracy of magnetic units. For instance, in this study, we used the dip coating method to fabricate tubes. When the tube diameter is reduced below 1 mm, this approach becomes unsuitable, as the surface tension of the liquid solution dominates, causing it to form spherical droplets rather than uniformly coating the surface of the mandrel (Supplementary Fig. 88 and Supplementary Video 21). Therefore, alternative fabrication methods need to be explored to produce tubes at smaller scales. Friction can be addressed by improving manufacturing precision and lubrication. Positional accuracy might be achieved through precise positional control systems, and their complexity increases as size decreases. For even smaller scales, alternative actuation mechanisms such as thermal, optical or chemical actuation might be considered to overcome the limitations of positional control systems. Identifying an appropriate and effective actuation method may require further exploration.

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

## Methods

### Fabrication

All the soft tubes were fabricated using the dip-coating moulding method. As shown in Supplementary Fig. 67a, initially, a commercial pin (steel test pin, ORION, DE) was fixed onto a flat plate to form a positive mould. Subsequently, this mould was immersed in a mixed polydimethylsiloxane (PDMS) (Sylgard 184, Dow Corning), with the base-to-curing agent mass ratio of 20:1. The coated mould was then positioned vertically and cured in an oven at 70 °C. Finally, by repeating the coating and curing processes (second and fourth steps), a soft tube of the desired thickness was obtained.

To fabricate a soft tube containing magnetic units, we first created a magnetic soft tube and then connected it with the previously fabricated soft tube. The magnetic tube was manufactured using a moulding method (Supplementary Fig. 67b). Initially, a positive mould was produced using 3D printing technology (Form 3B, Formlabs), with Clear V4 as the specified printing material, followed by plasma treatment to enhance its hydrophilicity. Subsequently, the positive mould underwent silanization treatment with trichloro(1H,1H,2H,2H-perfluorooctyl) silane (97%, Merck KgaA) to facilitate demoulding. The base and curing agent of Dragon Skin 10 (Smooth-On) were mixed in a 1:1 mass ratio, and the positive mould was then immersed in a container with this solution. After 10 min of degassing and curing at 90 °C for 1 h, the positive mould was removed to obtain a negative mould, which was then subjected to plasma treatment and silanization. Next, neodymium iron boron (NdFeB) microparticles (MQP-15-7, Magnequench) and PDMS (base and curing agent mixed in a 10:1 mass ratio) were blended in a certain proportion, which varied according to different requirements. The mixture was introduced into the negative mould containing a 3D-printed positive mould and then degassed in a vacuum chamber. After curing in an oven at 90 °C for 1 h, the mould was removed to obtain the magnetic tube. Finally, the magnetic tube was uniformly magnetized in a vibrating sample magnetometer (VSM) (EZ7, MicroSense) (Supplementary Fig. 70a) at 1.8 T. As shown in Supplementary Fig. 67c, the magnetic soft tube was then connected to the soft tube using PDMS, and the assembly was cured in an oven to obtain a soft tube containing magnetic units.

To facilitate processing and manufacturing, magnetic soft rods were used to replace the inner magnetic soft tubes in the composition of multilayer tubes. The fabrication method for the magnetic soft rod (Supplementary Fig. 68) is similar to that of the magnetic soft tube. The fabrication steps for the magnetic soft rod are simpler than those for the magnetic soft tube, with the main difference being the type of mould used.

The fabrication process for the spiral-deformed soft tube in configuration G in 1D magnetization reprogramming (Supplementary Fig. 15) involves the following steps. Initially, we created a magnetic soft tube using the dip-coating moulding method, as shown in Supplementary Fig. 67b. In the second step, the materials substituted for PDMS were a mixture of ecoflex (ECOFLEX 0050, KauPo), NdFeB microparticles (MQP-15-7, Magnequench) and platinum silicone cure retarder (SLO-JO/1, KauPo). The mixture was prepared with a mass ratio of retarder to ecoflex at 3%, and the ratio of ecoflex to microparticles was adjusted as per specific requirements. After obtaining the spiral-deformed soft tube, it was wrapped around a cylinder with a diameter of 12 mm and fixed in place (Supplementary Fig. 15b). It was then magnetized using a magnetic field of 1.8 T in the VSM, resulting in the desired soft tube. The fabrication of the magnetic and non-magnetic rods, which are paired with the spiral-deformed soft tube, followed the same procedure as that for producing soft rods and magnetic soft rods (Supplementary Fig. 68). For the magnetization of the magnetic rods paired with the spiral-deformed soft tube, a magnetic field opposite in direction to the spiral-deformed soft tube was used.

The method for producing the perforated sheet shown in 2D magnetization reprogramming was as follows (Supplementary Fig. 69a).

Initially, a 3D printed mould, comprising a base with fixed holes and square rods, was used. This mould underwent plasma and silane surface treatment. Subsequently, a mixture of ecoflex (ECOFLEX 0030, KauPo) and platinum silicone cure retarder (SLO-JO/1, KauPo) was prepared, maintaining a mass ratio of the retarder to ecoflex at 3%. The mould with this mixture was then placed in a vacuum chamber to remove air and, after 10 min, transferred to an oven to cure for 60 min. On completion, the mould was removed, yielding the perforated sheet. The square rod with magnetic units, intended to be paired with this perforated sheet, was fabricated in a manner similar to the magnetic soft rods (Supplementary Fig. 68). The primary distinction involved replacing cylindrical moulds with square ones.

To obtain a 3D box-shaped soft structure, multiple perforated sheets that we produced were connected. For this connection, unperforated sheets were fabricated using the same method as for the perforated sheets, with the key difference being the use of a mould without holes (Supplementary Fig. 69b). Two unperforated sheets and two perforated sheets were bonded and cured with ecoflex, resulting in the formation of the 3D box-shaped soft structure (Supplementary Fig. 69c). In this study, to enhance visibility, a red pigment (SILC-PIG, KauPo) was added to all 1D, 2D and 3D samples during fabrication, except for those containing magnetic particles. The samples with magnetic particles remained black because of the inherent colour of the magnetic material, which cannot be masked by the pigment.

### Experimental platforms

Two types of magnetic units were used in the experiments on the soft robots to magnetically actuate them. The first unit type comprises commercially purchased tiny permanent magnets with preset magnetization profiles. The second unit type includes soft elastomeric materials that are embedded with magnetic particles. We used the VSM to magnetize these materials at 1.8 T to achieve the required magnetization profiles (Supplementary Fig. 70a). The magnetic characterization of different materials mixed with varying proportions of magnetic particles was also measured using the VSM.

This study used two types of external magnetic fields for actuation: a uniform magnetic field and a rotating magnetic field. The uniform magnetic field was generated using the VSM, with its magnitude adjustable as required. In some experiments, different Halbach arrays were also used to generate uniform magnetic fields. As shown in Supplementary Fig. 70c, a Halbach array is mounted on a direct current (d.c.) motor shaft through connecting components. Given that the Halbach array can produce a uniform magnetic field, rotating the array results in the creation of a rotating magnetic field. By adjusting the rotational speed of the d.c. motor, the frequency of the rotating magnetic field can be precisely controlled.

To determine the elastic modulus of different materials, we used a universal testing system (5942, Instron) for tensile testing, as shown in Supplementary Fig. 70b. The test involves continuously applying a tensile load until the sample fractures. This process allows us to obtain the displacement and load applied during the stretching of the sample, thereby facilitating the calculation of the material's elastic modulus. Furthermore, this instrument was also used to measure the pulling force and friction of rods in soft tubes with varying curvatures.

### Modelling

This study used the pseudo-rigid-body method to analyse the deformation of soft tubes in magnetic fields. In this approach, the soft tube is represented as a series of rigid rods linked by hinges. These segments remain undeformed under external forces while the hinges undergo rotational motion. The elastic potential energy resulting from the deformation of the soft tube is assumed to be stored in these deforming hinges. As the soft tube deforms, the angles between the rigid rods change. By subdividing these segments, a curve composed of segments

at varying angles can be constructed, thereby approximating the deformation of the soft tube. This study applied the energy method, specifically the principle of minimum potential energy, to approximate the deformation of the soft tube.

To simplify and accurately analyse the deformation of soft tubes, this study opted for suspending the tubes. Similar force analysis processes apply to other placement methods, requiring only adjustments in the direction of the applied forces or the addition of corresponding forces. When a magnetic field is applied to the soft tube, it is subjected to both gravity and the magnetic field. The total potential energy of the system is composed of magnetic potential energy, elastic potential energy and gravitational potential energy. The total potential energy is expressed as

$$V = U_m + U_g + U_e \tag{1}$$

where $U_m$ represents the magnetic potential energy, $U_g$ represents the gravitational potential energy and $U_e$ represents the elastic potential energy of the system.

In this study, the soft tube was modelled as a structure consisting of $n$ rigid rods connected by $n − 1$ hinges (Supplementary Fig. 57), with $m$ of these rods possessing magnetic properties. Within a magnetic field, these magnetized rigid rods experience magnetic forces, and the total magnetic potential energy of these $m$ magnetic rods can be represented as

$$U_m = -\sum_{j=1}^{m} V_{mj} \tag{2}$$

Here, $V_{mj}$ represents the magnetic potential energy of a single rod, which can be calculated using

$$V_{mj} = \mathbf{m}_j \cdot \mathbf{B}_j \tag{3}$$

where $\mathbf{m}_j$ denotes the magnetic moment of the $j$th rod and $\mathbf{B}_j$ represents the magnetic field experienced by the $j$th rod. The magnetic moment of the $j$th rod can be obtained by

$$\mathbf{m}_j = m_j \mathbf{r}_j \tag{4}$$

where $m_j$ represents the magnitude of the magnetic moment of the $j$th rod and $\mathbf{r}_j$ is the unit direction vector of the $j$th rod. Therefore, the total magnetic potential energy of the system can be expressed as

$$U_m = -\sum_{j=1}^{m} m_j \mathbf{r}_j \cdot \mathbf{B}_j \tag{5}$$

As the soft tube is modelled as a combination of rigid rods and hinges, the elastic potential energy of the system is stored only in the hinges. The elastic potential energy in a hinge depends on the angle between adjacent rods. Thus, the total elastic potential energy of the system can be calculated by

$$U_e = \frac{2n}{L} \sum_{i=1}^{n} E_i I_i \theta_i^2 \tag{6}$$

where $E_i$ represents the modulus of elasticity of the $i$th rod, $I_i$ is the moment of inertia of the cross-section of the $i$th rod and $\theta_i$ is the angle between the $(i + 1)$th rod and the $i$th rod.

To facilitate calculations, this study assigned corresponding vectors to each rod. The initial vector of the $i$th rod is denoted as $\mathbf{p}_i^i$, and on the deformation of the soft tube under the influence of external forces, the final vector of the rod becomes $\mathbf{p}_i^f$. Assuming that the rigid rods in the pseudo-rigid body model have a uniformly distributed mass, the centre of gravity of each rod coincides with its geometric centre.

Assuming the mass of the $i$th rod is $m_i^m$, the gravitational potential energy of the system can be represented as

$$U_g = -\frac{1}{2} \sum_{i=1}^{n} m_i^m (\mathbf{p}_i^f - \mathbf{p}_i^i) \cdot \mathbf{g} \tag{7}$$

where $\mathbf{g}$ is the acceleration due to gravity.

Through the aforementioned analysis, we derived the magnetic potential energy, elastic potential energy and gravitational potential energy. According to equation (1) and using the principle of minimum potential energy, we can determine the static equilibrium configuration by finding the local minima of the potential energy. This allows us to obtain the angles between the rods, denoted as $\theta_i$. The potential energy equation can be solved using the fmincon function in MATLAB (R2022a, MathWorks), which is a solver for finding the minimum of a constrained nonlinear multivariable function.

To measure the deformation of the soft tube, we selected two parameters: the terminal bending angle and the minimum radius of curvature in the bending area (Supplementary Fig. 29). If there are multiple bending areas, then there will be multiple minimum radii of curvature. Based on the solved data $\theta_i$, the terminal bending angle and minimum radius of curvature were obtained through MATLAB. More details regarding the modelling and calculations are provided in Supplementary Note 2, and the theoretical results obtained from the above calculations are presented in Supplementary Fig. 27, in which they are compared with the experimental results.

In the deformation analysis of 2D sheets or 3D bodies, the aforementioned method of calculations using theoretical formulas becomes excessively complex. Alternatively, we used finite element simulation using ABAQUS, based on the method provided in ref. 22. We obtained the simulation results for the 2D sheets and compared them with experimental results (Supplementary Fig. 28).

## Preparation for experiments

The avoidance object model with cylinders was produced using 3D printing (clear V4, Formlabs). The cylinder has a diameter of 12 mm and a height of 20 m. It was positioned on a flat surface to form the required model. Specific geometric parameters of the model are shown in Supplementary Figs. 34 and 35. The vascular model was also fabricated using 3D printing and subsequently underwent surface treatment (SCS PDS 2010 Labcoter, Specialty Coating Systems) with Parylene C. Specific geometric parameters of the vascular model are provided in Supplementary Fig. 37. The base plate for securing the cilia was crafted from acrylic using laser processing to create an array of small holes on its surface, into which the cilia were then fixed. The ciliary paddles at the tips of the cilia were also produced using 3D printing technology (clear V4, Formlabs) and the specific geometric parameters are detailed in Supplementary Fig. 53. The tissue model was fabricated using 3D printing with the soft material Elastic 50 A (Formlabs) selected to mimic the soft properties of tissue. The object model was also created through 3D printing, using the material clear V4.

## Data availability

All data that support the findings of this study are provided in the main text and the Supplementary Information.

## Code availability

The MATLAB codes used in this study are available from the corresponding author upon request.

**Acknowledgements** This work was funded by the Max Planck Society, European Research Council Advanced Grant SoMMoR project (grant no. 834531) and the German Research Foundation Soft Material Robotic Systems (SPP 2100) Program (grant no. 2197/3-1). X.B. and R.Z. thank the Alexander von Humboldt Foundation for financial support. J.Z. and M.S. thank

the Max Planck Queensland Center (MPQC) for the Materials Science of Extracellular Matrices for financial support.

**Author contributions** X.B., F.W., J.Z. and M.S. conceived the idea and designed the research. X.B. carried out the implementation of the design, system tuning and theoretical analysis. X.B., F.W. and J.Z. conducted the experiments and analysed the data. X.B. carried out the simulations of deformation. F.W. performed the simulations of multi-magnetic field interference. M.S. supervised the research. X.B., F.W., J.Z. and M.S. wrote the paper. M.L., S.Z., Z.R., J.L., Y.Y., W.K., R.Z., Z.L. and T.W. contributed to the discussions.

**Funding** Open access funding provided by Max Planck Society.

**Competing interests** X.B., M.S., F.W. and J.Z. have filed a patent application (application no. EP25178039.1) covering the core concept and method described in this paper. All other authors declare no competing interests.

**Additional information**
**Correspondence and requests for materials** should be addressed to Metin Sitti.
