## [Peer Review File · Nature]

Real-Time In-Situ Magnetization Reprogramming for Soft Robotics

Corresponding Author: Professor Metin Sitti

Version 0:

Reviewer comments:

Referee #1

(Remarks to the Author)

This manuscript presents a novel method for real-time and in situ reprogramming the magnetic configuration in magnetic soft robots. By nesting multiple tubes and rods with different magnetizations and adjusting their relative positions, the overall net magnetization of the entire structure is effectively manipulated. This approach modulates shape morphing without the need to change the external magnetic field. Furthermore, the authors demonstrate four potential applications: contact-free navigation, programmable cilia arrays, coordinated operations, and shape-adaptable soft grippers. From a fundamental standpoint, I find the authors' idea of altering shapes by adjusting magnetization—rather than tuning the magnetic field as usual—to be very creative. The work is certainly well-executed, with comprehensive experimental results reported. I believe this manuscript will interest not only the magnetic soft robot community but also the broader readership of Nature. However, I feel that the following issues need to be addressed before acceptance:

1. The current forms of Figures 1 and 2 are quite difficult to understand. The analogy made in Figure 1a is misleading. In phototropism, sunlight induces the relocation of auxin in the stems of a sunflower, causing cells on the shady side to elongate and thus pushing the plant toward the sun. In the authors' reprogramming strategy, however, the magnetic field—depicted similarly to sunlight in the figure—does not induce the relocation of the magnetic units in the tubes. Instead, the magnetic units are moved by external forces (I guess, since this is not clearly depicted in the manuscript) through pulling or pushing the tubes or rods containing them. I suggest the authors improve this schematic to make the concept easier for readers to understand.

2. Figure 2 does not sufficiently illustrate the relative motion of the nested tubes. I recommend enhancing the schematic to emphasize that the parameters d_{Ri+1} are tunable. Providing a series of schematics for the experiment in Figure 2a(iii), highlighting the relative positions of tubes R1 and R2, would be very helpful.

3. In Figure 3, the authors demonstrate contact-free object navigation with the bending portion anchored during catheter motion. While this is core to achieving contact-free navigation—requiring more anchored bends to avoid contact with additional obstacles—the concept seems limited to planar operations. Is it possible to achieve 3D contact-free object navigation with multiple anchored bends arranged in three dimensions? This is important for catheter applications that require 3D navigation within blood vessels and various human body cavities.

4. For the coordinated operation demonstration in Figure 5, it appears the concept is restricted to planar operations because all shape transformations under the same magnetic field occur in a plane parallel to the working environment. It seems challenging to induce motions out of this plane. Please properly evaluate the limitations of this concept.

5. I suggest the authors provide a clear definition of the term "magnetic unit," which is used frequently throughout the manuscript. For example, on page 4, line 109, the authors mention "even an infinite number of magnetic units" for Figure 1B(i), which is confusing.

6. While the manuscript demonstrates the experimental setup for magnetizing the samples and applying magnetic fields, it lacks details on controlling the relative motion of the nested tubes or rods. Are they manipulated manually, or is there a

motorized control system involved?

7. Two important recent works on magnetic continuum robots for contact-free navigation—*Nat. Commun.* 15, 3759 (2024) and *Adv. Funct. Mater.* 2024, 2412543—have not been properly cited. Including these references would strengthen the manuscript's context within the field.

8. In Figure 3B, there are two sections labeled (ii). One of them should be labeled (iii) to maintain proper sequencing.

Referee #2

(Remarks to the Author)

Actuators based on concentric tubes with different magnetization directions in each layer are reported, where axial translation of the tubular layers dynamically alters the total magnetization profile, thus reprogramming the response of the tube. Single tubes are envisioned as surgical catheters that can avoid obstacles while being inserted, and multiple tubes are shown working together in a way that could mimic multiple catheters. Arrays of tubes are demonstrated as dynamically tunable cilia that can pump fluids. Forming bundles of multiple tubes or connecting them with non-responsive elastomer layers demonstrates additional capabilities, such as morphing 2D sheets, boxes, and grabbers. This work is elegant, creative, and comprehensive.

Despite these strengths, in my assessment, this work does not rise to the level of novelty and impact that are needed for publication in *Nature*, though it comes close. The second paragraph of the Introduction describes the challenge and the need to reprogram the magnetization profile, while referencing approaches that reorient magnetic domains. This work reprograms the total magnetization another way, which is innovative but has some significant limitations: The structures are tethered to adjust the positions of the tubes, and reprogramming is accomplished through partial cancellation of the magnetization, creating a tradeoff between programmability and the strength of the response. Reprogramming through mechanical motion of the tubes will restrict scaling to much smaller sizes. There are also practical limitations to the number of layers in a tube, which will ultimately limit the behaviors attainable through reprogramming. For example, the length of the tubes reported in this work is about 50 mm, but the more clinically relevant case of a catheter with a length of 900 mm is given. It may be possible to avoid a couple of obstacles, but there is no clear approach for avoiding many obstacles.

Additional comments:

1. What causes the orange color of these structures?
2. Why is the magnetometry measurement in Fig. S65a asymmetrical?
3. Is there an error in Fig. S66B(vi)? The negative mold shows narrow tips that would be completely filled, but they are missing two panels below in (viii).
4. For avoiding obstacles and coordinated operation of multiple catheters, it seems like the anatomy would need to be known in advance to appropriately magnetize the tubes and plan the motion. Is that correct, or is it possible with this approach to navigate in an unknown environment, based only on imaging near the tip of the catheter?

Referee #3

(Remarks to the Author)

Metin Sitti and his colleagues report core-shell structured multiple-tubes using movable magnetic units as core materials. By doing so, real time in-situ reprogramming of magnetization and magnetic actuations were enabled. The research team then demonstrated potential applications in biomedical fields. The demonstration of biomedical applications with controllable tube configurations is impressive due to its potential for practical applications. Conclusions are well-supported by the extensive amount of experimental data for parameter studies with theoretical analysis.

However, I have mixed feelings about reading this manuscript. While the demonstration is neat, basic idea has been shown several years ago. The authors stated that "The real-time in-situ magnetization reprogramming proposed in this paper is achieved by dynamically altering the positions of magnetic units". Core-shell structure with different position of magnetic cores have been shown in 2020 *Advanced Materials* (*Adv. Mater.* 2020, 32, 2001879) by Zhengzhi Wang. Parameter studies have shown in 2020 *Extreme Mechanics Letters* (*Extreme Mechanics Letters* 38 (2020) 100734). The same author reported reprogramming of magnetic actuation by dynamic re-location of magnetic particles in core-shell structured micropillars in 2021 *ACS Nano* (*ACS Nano* 2021, 15, 4747–4758). The previous study also showed writing and rewriting of area-selective patterned actuation of micropillars (or cilia for terminology in the manuscript). While the submitted manuscript is showing magnetic reprogramming not just location of magnetic particles, this was possible due to cm scale of tubes. The 2021 *ACS Nano* paper showed writing and rewriting of 10 micron-sized pillar arrays. In particular, see their Figure 4, SI Video 10 and SI Video 11 for patterning of micropillar actuations.

Surprisingly, none of the aforementioned previous papers are cited in this manuscript although this manuscript cited 92 references. Hence, if this is not intentional, I believe that the authors are totally unaware of these previous reports. While this manuscript utilizes the core-shell tubes for faster relocation of core tube, this is possible due to the larger tube size. The same concept would not be available for micron scale. The 2021 *ACS Nano* shows relocation of magnetic units within 10-20 s.

Due to the presence of previous reports on the key concepts including smaller scale demonstration, I cannot support for publication of this manuscript in Nature. I recommend authors to transfer this manuscript to Nature Materials or Nature Communications.

Version 1:

Reviewer comments:

Referee #1

(Remarks to the Author)

The resubmitted manuscript has well addressed the concerns raised in my previous review. I am satisfied with the revisions and consider the work suitable for publication in Nature.

Referee #2

(Remarks to the Author)

I appreciate the comprehensive responses to the reviews. The authors have addressed my concerns and make a persuasive case for the suitability of this work in Nature. The mechanism of dynamic reprogramming is distinct from other work and impactful for applications. The additional demonstrations added in the revision highlight this potential.

Referee #3

(Remarks to the Author)

The main claim of this manuscript is gradient induced programming of local deformation and reprogramming of magnetic actuation modes. As I pointed out from last review, there are three very relevant but uncited papers.

Ref-1: Hybrid Magnetic Micropillar Arrays for Programmable Actuation, *Advanced Materials*, 2020, 32,2001879.

Ref-2: Heterogeneous magnetic micropillars for regulated bending actuation, *Extreme Mechanics Letters*, 2020, 38, 100734.

Ref-3: Core-Shell Magnetic Micropillars for Reprogrammable Actuation, *ACS, Nano* 2021, 15, 4747-4758.

Although Ref-1 and Ref-2 are not dealing with magnetic reprogramming, varying deformation behaviors in micropillars were generated by moving magnetic nanoparticles under applied magnetic fields.

"We found that all three studies share a similar underlying concept, namely achieving varying deformation behaviors in micropillars by generating different concentration distributions of magnetic nanoparticles under applied magnetic fields. However, the three studies differ in their research focus: Ref-1 and Ref-2 investigate magnetization programming techniques, while Ref-3 explores magnetization reprogramming techniques."

Hence, the authors admit that the novelty of this manuscript is not about varying deformation behaviors based on mechanical stiffness gradient to program local deformation, but about magnetization reprogramming. The method we proposed in our study falls into the category of magnetization reprogramming."

"Given that Ref-1, Ref-2, and Ref-3 present methods similar to each other, and only Ref3 aligns with our approach of magnetization reprogramming, we will primarily focus our comparative analysis on Ref-3. The comparative analysis will be presented in terms of the following four aspects: technical strategy, resulting performance, application scope, and impact on the current field."

Also, authors admit that the magnetization reprogramming has been reported before.

"Our proposed method achieves the repositioning of magnetic units directly through the internal movement of soft robotic components themselves, subsequently enabling deformation under external magnetic fields."

The Ref-3 repositioned the location of magnetic nanoparticles within micropillars. Meanwhile, this manuscript utilized large millimeter scale tubes and that's why the control of internal movement is allowed. If authors can achieve the same thing with micropillars, I will definitely say yes to this manuscript for Nature. However, internal movement of magnetic nanoparticles within micropillar is reported in 2021 *ACS Nano*. The 2021 *ACS Nano* reported magnetization reprogramming and resultant change in actuation behaviors due to stiffness gradient. The concept-wise, the idea is quite similar and the previous paper was published 4 years ago. In addition, the size of actuator was much smaller. In case of micron-scale tubes, it will be significantly more difficult to achieve the same idea due to the fabrication difficulty, difficult control, large friction, and adhesion between tubes after actuation.

"In Ref-3, the repositioning of nanoparticles must first be accomplished using a specialized device (a magnetic needle) before any deformation can occur under external magnetic fields"

This is to remotely control the small scale components (nanoparticles). Macro-scale parts would not need remote control.

"In our proposed method, however, repositioning is achieved through the relative motion of the soft robot's components, entirely independent of the external magnetic field. The repositioning operation and magnetic actuation can occur simultaneously, thus enabling real-time magnetization reprogramming."

I do agree with the important novelty and first demonstration of this new concept in macroscale. Authors showed beautiful demonstrations and extensively investigated the phenomena. Other than the novelty issue, every component of the manuscript is at top-notch. However, I still don't agree with the publication of this manuscript in Nature due to the previous

publication (2021 ACS Nano) for important concept on magnetization reprogramming and mechanical stiffness-induced change in deformation of micron-scale actuators with control of position of nanoscale components. Hence, I again recommend transferring this manuscript to Nature Materials or Nature Communications.

Referee #4

(Remarks to the Author)

The manuscript entitled “Real-Time In-Situ Magnetization Reprogramming for Soft Robotics” reports a method for recombination of magnetic units to achieve varied magnetization profiles, thus enabling in-situ shape manipulation of magnetic soft robots without changing external magnetic field. Comprehensive application scenarios are demonstrated and well-supported by experimental results.

Overall, the proposed method endows the magnetic soft robot with impressive deformation capabilities under simple magnetic field control and the manuscript is also well-organized. However, some inherent limitations listed as follows constrain the novelty, flexibility, and application potential of the proposed method, which makes the work not yet meet the standards required for publication in Nature.

1. Unlike other magnetization reprogramming methods, the method proposed in this manuscript does not allow for arbitrary control of the magnetization direction at a specific point/region. Instead, the magnetization can only be adjusted to a limited extent by combining multiple magnetized units. As such, this method might be more accurately described as magnetization recombination or redistribution, rather than fully magnetization reprogramming. Once the number and magnetization directions of the magnetic units in a given configuration are settled, all possible deformation patterns under a constant magnetic field are essentially determined.
2. In comparison to other magnetization reprogramming methods, the method presented in this work may compromise two key advantages typically associated with magnetic soft robots: untethered control and ease of miniaturization. This could substantially limit the practical applicability of robots based on this approach. The reconfiguration of magnetic units appears to require the use of many tethered actuators, which may result in a relatively bulky system. In this context, it may be worth considering whether magnetic actuation remains the most appropriate choice, especially when alternative tethered actuation strategies could potentially offer higher precision and more degrees of freedom.
3. The design space of the proposed method is directly tied to the number of magnetic units, which in turn determines the number of additional actuators required. Moreover, achieving higher degrees of freedom during the recombination of magnetic units would necessitate a significant increase in the number of actuators. For 1D configurations, the actuator count may still be manageable—as demonstrated by the authors using multiple motors. However, when higher resolution is needed, or when the configuration becomes 2D or 3D, the number of actuators required could grow substantially, leading to a highly complex actuation and control system. This would greatly limit the practical application of the method. For instance, in the manuscript, all reconfigurations beyond 1D are carried out manually.
4. The fabrication and inverse design of the proposed robot are also non-trivial. Considering that multiple magnetic units need to be integrated, manufacturing becomes particularly challenging when the robot size is small. Furthermore, in complex and unpredictable environments, it is difficult to design a single configuration that can adapt to diverse scenarios. This poses significant challenges in determining the appropriate number of magnetic units, their magnetization directions, and the corresponding actuation modes.

Response to Reviewer Comments

We thank all reviewers for their insightful comments and constructive feedback. Below, we replied comments point-by-point, where the reviewer comments are in blue, our replies are in black, and changes in the manuscript are highlighted in yellow.

To facilitate a quick understanding of the modifications and additions made in our manuscript, we provide the following detailed summary:

Changes made in the original manuscript:

Figures:

- Fig. 1, Fig. 2, Fig. 3
- Fig. S61, Fig. S65, Fig. S66

Manuscript:

- All changes are highlighted in yellow.

Notes:

- Note S3 (merged into Note S9, and the original Note S3 was deleted)
- Note S6

New Additions:

Figures:

- new Figs. S70–S95

Notes:

- new Notes S7–S11

Movies:

- new movies S16–S21

References:

- new references 35, 63, and 64 in the manuscript
- new references 6 and 7 in the supplementary materials

Please note that, for ease of reference, we have **maintained the original numbering of figures and notes** from the previously submitted manuscript. Newly added content follows the original numbering sequence. For example, since the original Note S3 has been integrated into the newly added Note S9, Note S3 is intentionally absent in the revised manuscript to preserve numbering consistency.

Referee #1:

Overall Assessment:

This manuscript presents a novel method for real-time and in situ reprogramming the magnetic configuration in magnetic soft robots. By nesting multiple tubes and rods with different magnetizations and adjusting their relative positions, the overall net magnetization of the entire structure is effectively manipulated. This approach modulates shape morphing without the need to change the external magnetic field. Furthermore, the authors demonstrate four potential applications: contact-free navigation, programmable cilia arrays, coordinated operations, and shape-adaptable soft grippers. From a fundamental standpoint, I find the authors' idea of altering shapes by adjusting magnetization—rather than tuning the magnetic field as usual—to be very creative. The work is certainly well-executed, with comprehensive experimental results reported. I believe this manuscript will interest not only the magnetic soft robot community but also the broader readership of Nature. However, I feel that the following issues need to be addressed before acceptance.

Response:

We thank the Reviewer for thorough evaluation of our manuscript and the insightful suggestions. Our point-by-point responses to each comment are provided below.

Comment 1:

The current forms of Figures 1 and 2 are quite difficult to understand. The analogy made in Figure 1a is misleading. In phototropism, sunlight induces the relocation of auxin in the stems of a sunflower, causing cells on the shady side to elongate and thus pushing the plant toward the sun. In the authors' reprogramming strategy, however, the magnetic field—depicted similarly to sunlight in the figure—does not induce the relocation of the magnetic units in the tubes. Instead, the magnetic units are moved by external forces (I guess, since this is not clearly depicted in the manuscript) through pulling or pushing the tubes or rods containing them. I suggest the authors improve this schematic to make the concept easier for readers to understand.

Response:

We greatly appreciate the Reviewer's careful examination of our figures and the constructive suggestions. As noted, Figs. 1 and 2 were not sufficiently clear. We have revised them accordingly, and the detailed changes and explanations are outlined below.

Our proposed reprogramming strategy was indeed inspired by sunflowers. However, the emphasis in our original description may have been misleading. As the Reviewer rightly noted, the overall mechanism of phototropism in sunflowers is not entirely comparable to the mechanism underlying our proposed method—particularly because the roles of sunlight and magnetic fields are fundamentally different. Therefore,

overemphasizing phototropism may have led to a misinterpretation. We believe the fundamental principles are morphologically captured in both systems: deformation is achieved by spatially redistributing internal units, which then induces downstream mechanisms. In the case of sunflowers, the redistribution of auxin leads to differential growth rates across different regions, resulting in deformation; in our proposed method, the repositioning of internal magnetic units leads to spatial variation in the magnetic forces or torques experienced under a magnetic field, thereby producing deformation.

Given that a similar core mechanism is fundamentally shared by both systems and that our work was indeed inspired by sunflowers, we believe it is rightfully captured by the sunflower-inspired concept. In the revised manuscript, to better emphasize this similarity, we modified Fig. 1A. Specifically, we added a schematic representation of auxin transporters responsible for auxin relocation, as well as magnetic unit carriers responsible for repositioning the magnetic units. The focus is placed on the idea that both systems achieve reconfiguration of their core functional units—auxin and magnetic units, respectively—through the transport or manipulation by associated carriers (i.e., auxin transporters or magnetic unit carriers). Moreover, we clarified this point in the figure caption and manuscript.

For the revisions to Fig. 2, please see our response to Comment 2.

The revised description in the manuscript is in lines 77-91 as:

We observe that sunflowers in nature achieve adaptive deformation by redistributing internal auxin in response to changes in solar position (Fig. 1A). Auxin transporters actively reposition auxin, altering its spatial distribution, thus inducing variations in local growth rates and resulting in differential deformation. Inspired by this biological phenomenon, we propose a real-time in-situ magnetization reprogramming approach that achieves deformation by redistributing internal magnetic units through the movement of magnetic unit carriers. Specifically, changes in the positions of magnetic unit carriers relocate magnetic units, altering their spatial distribution (manifesting as distinct magnetization profiles), thus inducing variations in forces and torques experienced under external magnetic fields, resulting in differential deformation. Both the sunflower and our proposed method achieve differential deformation through the redistribution of their respective internal elements—auxin in the sunflower and magnetic units in our method. For comparison, auxin transporters in sunflowers are driven internally through active biological mechanisms, whereas magnetic unit carriers in the proposed method are externally controlled by various actuators or stimuli capable of inducing different modes of motion.

Also, the revised Fig. 1 and its caption are as follows:

Fig. 1. Sunflower-inspired real-time in-situ magnetization reprogramming of magnetic soft robots. (A) Adaptive response of sunflowers mediated by internal auxin redistribution, inspiring a similar mechanism for real-time in-situ magnetization reprogramming. Both systems achieve differential deformation by redistributing their internal elements—auxin in sunflowers and magnetic units in our proposed method. Sunflowers induce auxin transporters in response to illumination to transport auxin, causing variations in auxin distribution, whereas the proposed method induces magnetic unit carriers according to operational requirements to move magnetic units, causing variations in magnetic unit position. Parallel to the auxin transporters in sunflowers, which are internally regulated by biochemical processes, the magnetic unit carriers in the proposed method are controlled by external actuation. (B) Under a uniform magnetic field, several demonstrations of deformation are achieved through the proposed real-time in-situ magnetization reprogramming, ranging from (i and ii) one-dimensional (1D) tubes to (ii) two-dimensional (2D) sheets, and (iii) progressing to three-dimensional (3D) box-shaped structures, with transformations occurring at different moments (movie S1-S2). The tops of the objects in (i-iv) are suspended, with (i-iii) shown in the front view and (iv) depicted in the bottom view. In (i), the tube transitions from a straight line into a helix under the same uniform magnetic field; in (ii), the tube achieves deformations to the left and right under the same uniform magnetic field. In (i), the tube's color was digitally adjusted from black to red for enhanced visualization, whereas the objects in (ii-iv), originally red, were left unmodified. Scale bar: 5mm.

Comment 2:

Figure 2 does not sufficiently illustrate the relative motion of the nested tubes. I recommend enhancing the schematic to emphasize that the parameters d_{Ri+1} are

tunable. Providing a series of schematics for the experiment in Figure 2A(iii), highlighting the relative positions of tubes R1 and R2, would be very helpful.

Response:

We greatly appreciate the Reviewer’s careful review of our figures and the insightful suggestions. A new subfigure (Fig. 2A(ii)) is added to illustrate the relative motion between the nested tubes, highlighting that the parameter d_{Ri+1} is tunable. We also included a schematic of the tube configuration in the original Fig. 2A(iii), now updated as Fig. 2A(iv). The relevant explanatory text has also been incorporated into the manuscript to improve clarity regarding the working principle.

The added description in the manuscript is in lines 140-141 as:

The relative positions between tubes (d_{Ri+1}) are tunable according to requirements (Fig. 2A(ii)).

Revised Fig. 2A and its caption are shown below:

Fig. 2. Illustration of the real-time in-situ magnetization reprogramming method. (A) 1D magnetization reprogramming. (i) Multiple tubes, each embedded with a magnetic unit, undergo a nested assembly to form an integrated tube. R_j represents magnetic unit j in the tube, where the blue arrow indicates the magnetization direction of the magnetic unit. d_{Ri+1} means the distance between magnetic units i and $i+1$. (ii) The reconfiguration is achieved through the relative movement between tubes, resulting in different magnetization profiles. The parameter d_{Ri+1} can be tuned to meet requirements. (iii) Magnetization expansion of a single tube. In A(i),

Comment 3: In Figure 3, the authors demonstrate contact-free object navigation with the bending portion anchored during catheter motion. While this is core to achieving contact-free navigation—requiring more anchored bends to avoid contact with additional obstacles—the concept seems limited to planar operations. Is it possible to achieve 3D contact-free object navigation with multiple anchored bends arranged in three dimensions? This is important for catheter applications that require 3D navigation within blood vessels and various human body cavities.

Response:

Thanks for this constructive feedback. In response to the comments on 3D navigation, we included additional design methodologies, analyses, discussions, and experimental demonstrations in the revised manuscript. The details of these additions are as follows:

(1) To evaluate the feasibility of 3D navigation, we theoretically demonstrated that the proposed method is capable of achieving controlled 3D motion under a constant magnetic field. The detailed analysis is provided in Note S7.

(2) We employed a Halbach array to simulate a single static magnetic field. Within this field, we demonstrated 3D operation and the ability of the tube to switch between different operating modes. Please see movie S16.

(3) We fabricated two different 3D obstacle models. Under a single magnetic field, we demonstrated that with two distinct tube configurations, the tube can smoothly bypass both types of obstacles through 3D deformation. The details are provided in Figs. S79-S82 and movie S17.

The 3D navigation shown in movie S17 is accomplished under a single-direction magnetic field. However, achieving traversal over a greater number of and more complex obstacles requires a larger magnetic field region and more complex magnetization profiles. Generating such large-scale magnetic fields is constrained by fabrication, cost, and other practical limitations. To address this, we proposed a multi-magnetic field control method, which utilizes multiple small-scale magnetic fields as an alternative approach (Section 4.1.3 in Note S9). In addition, to simplify the design of magnetization profiles, we introduced a multi-bend deformation design method (Section 4.2 in Note S9). Since we currently do not have appropriate electromagnetic coils for implementing the multi-magnetic field control method, we fabricated two Halbach arrays to substitute for the large-scale magnetic field (Fig. S84). Accordingly, we designed a customized tube configuration based on the multi-bend deformation design method (Fig. S83). In movie S18, the tube demonstrates planar bending under the first magnetic field (similar to 2D obstacle crossing shown in movies S4-S5). Under the second magnetic field, the tube performs 3D navigation (similar to the 3D obstacle traversal in movie S17), and further exhibits the capability of switching between different operating modes. With the addition of more magnetic fields, the system is capable of performing a greater number of bends, thereby enabling it to traverse more diverse obstacles. Since we are unable to control the magnetic field strength of the

Halbach array, the operation in movie S18 was implemented with the following two limitations:

1) The tube could not achieve smooth deformation as it did in the VSM setup. Because the magnetic field generated by the Halbach array was constantly present and could not be gradually ramped up, the tube began to bend even before fully entering the magnetic field region.

2) We were unable to demonstrate real-time obstacle avoidance. In the VSM system, the tube was shown to gradually bypass obstacles without contact, enabled by the ability to incrementally increase and decrease the magnetic field strength. However, because the Halbach array does not allow for field strength modulation, we could not demonstrate how the tube navigated around obstacles in real time. As an alternative, we only presented a proof-of-concept demonstration showing that the tube can achieve multiple bending deformations and traverse more obstacles (without actual obstacles being placed in the demonstration) in move S18. We believe that if the magnetic field strength of the Halbach array is tunable, it will enable multiple deformation modes and complex obstacle traversal.

To address the issue of magnetic field control, we have proposed specific solutions as a continuation of this research (Section 4.2.2 in Note S9). The first solution involves designing a mechanism to adjust the diameter of the circular ring arrangement of permanent magnets in the Halbach array. By changing the ring diameter, the magnetic field strength can be modulated (see Fig. S89 for the relationship between magnetic field strength and distribution diameter). The second solution is to replace the Halbach array with electromagnetic coils to enable tunable fields. Furthermore, to lay the foundation for future research, we conducted simulations on potential interference issues arising from spatial arrangements of multiple magnetic fields (Note S11).

(4) Given the operational length constraints in certain applications, such as navigation within blood vessels, we have proposed a specific design strategy to enable long-range operation, as detailed in Section 4.1 of Note S9. We subdivided the operating region into three distinct zones (pages 1-2 in Note S9): *Magnetic Interaction Zone* (where the magnetic field is present), *Actuation Zone* (where the tube is actively driven to achieve relative motions), and *Feeding Zone* (connects the Magnetic Interaction Zone and Actuation Zone). Consequently, the total length of the tube that can be operated is given by the sum of the tube lengths within the Magnetic Interaction Zone and the Feeding Zone.

Based on the aforementioned zone division, we categorized the operations into three types for clarity of explanation: long-range operation with a large Magnetic Interaction Length, long-range operation with a large Feeding Length, and long-range operation with a large Magnetic Interaction Length and large Feeding Length, where the third type represents a combination of the first two. Regarding the long-range operation with a large Magnetic Interaction Length, we demonstrated that a tube with a length of 305 mm can be manipulated while achieving multiple bending deformations (Figs. S83-S84,

movie S18). This length can be further increased with enhanced stiffness of tubes. For the long-range operation with a large Feeding Length, we integrated commercial catheters for manipulation, demonstrating an achievable operational length of at least 1240 mm (**Fig. S85, movie S19**). Since there is no magnetic field in the Feeding Zone in this type of operation, the tube does not undergo deformation in this region. Therefore, we employed a commercially available catheter to replace the high-stiffness tube in the Feeding Zone. We believe that the aforementioned operational distances essentially satisfy most application requirements. Furthermore, if necessary, longer distances can be achieved by summing the Magnetic Interaction Length and the Feeding Length.

Additionally, in **Section 4.1** of **Note S9**, we proposed a segmented design method and a multi-magnetic field control method for addressing design challenges associated with long-range operations. Due to the extensive details involved in these methods, we refrain from elaborating further here; please refer to **Section 4.1** of **Note S9** for a detailed discussion.

Furthermore, to provide a framework for future implementations, we provided design strategies tailored to different application scenarios in the revised manuscript. These include design options, optimization methods, parameter effects on performance, magnetization profile design, magnetic field configuration, and related analyses, and so on. The relevant content is detailed in **Notes S6, S9, S10, and S11**. As the amount of supplementary information is substantial, we cannot include all of it in this response. We kindly refer the Reviewer to the attached supplementary materials for further details.

We have added a description of the force-torque space and the additional work we have done in **lines 218-221** and **lines 228-231** as:

In addition, we theoretically demonstrated that the force-torque space achievable by existing methods is strictly a subset of that enabled by our proposed method when combined with a tunable magnetic field, highlighting the broader actuation capability of our approach (**Fig. S70, Note S8**).

Additionally, we provide the theoretical analysis and simulation calculations, manufacturing details, various potential analyses, magnetization profile design method, and specific design schemes and tuning strategies for various application scenarios in the “Materials and Methods” section and **Notes S1-S11**.

To give more descriptions of the operational length and other considerations, we added more text in **lines 342-358** as:

To validate the capability of the proposed method for achieving 3D contact-free object navigation, we fabricated two different 3D objects to avoid contact (**Figs. S80-S81**). By designing different magnetization profiles, we achieved contact-free navigation through these 3D objects under a single magnetic field (**Fig. 3H, Figs. S79 and S82, movie S17**). In future practical applications, there will likely be increased demands on the operational length, the number of bends, the curvature, and the number of layers. To

address these specific challenges, we proposed four new methods—namely, a segmented design method, a multi-magnetic field control method, a multi-bend deformation design method, and a zone-based tube nesting design method—targeting long-range operation, multi-bend deformation, high-curvature deformation, and multi-nested tube operation, respectively. For example, regarding the long-range operation, we demonstrated that a tube with a length of 305 mm could be manipulated while achieving multiple bending deformations (Figs. S83-S84, movie S18). This length can be further increased with enhanced stiffness of tubes. We integrated commercial catheters for manipulation, demonstrating an achievable operational length of at least 1240 mm (Fig. S85, movie S19). We believe that the aforementioned operational distances essentially satisfy most application requirements. The detailed designs, tuning strategies, and related discussions of these methods are presented in Section 4 of Note S9.

Comment 4: For the coordinated operation demonstration in Figure 5, it appears the concept is restricted to planar operations because all shape transformations under the same magnetic field occur in a plane parallel to the working environment. It seems challenging to induce motions out of this plane. Please properly evaluate the limitations of this concept.

Response:

We thank the Reviewer for this constructive comment. We regret that a demonstration was not included in Fig. 5. As explained in our response to Comment 3, the proposed method is not restricted to planar operations—it is capable of enabling various 3D operations even under a uniaxial static magnetic field with an appropriate magnetization profile. In Note S7, we theoretically demonstrated that 3D operations can be achieved using the proposed approach without altering the external magnetic field.

Although we did not explicitly present such 3D operations in Fig. 5, they have already been demonstrated in Fig. 1B(i), Fig. S14D, and movie S1, corresponding to Configuration G in the 1D magnetization reprogramming strategy. Specifically, when d_{R12} is 0 mm, the tube remains straight under a constant magnetic field directed to the right; with the magnetic field unchanged, varying d_{R12} induces different helical deformations of the tube. This scenario corresponds to the coordinated multi-instrument operation in Fig. 5, where the tube can modulate its own configuration to be selectively influenced by or shielded from the external magnetic field, thereby enabling various real-time deformation modes under an unchanging field. Thus, our proposed method can achieve 3D operations within the coordinated multi-instrument operation framework by utilizing Configuration G in the 1D magnetization reprogramming scheme. The complexity of the 3D deformation can be precisely controlled through the appropriate design of the magnetization profile associated with this configuration.

Moreover, by combining this strategy with the multi-magnetic field control method and the multi-bend deformation design method proposed in the revised manuscript, even

more complex 3D operations can be achieved (see also our response to Comment 3 for details).

To further illustrate the concept of coordinated multi-instrument operation, we have added corresponding descriptions in lines 579-590 as:

The above design utilizes the magnetic neutralization feature and Configuration B of the 1D magnetization reprogramming. Since the magnetic field direction remains constant, the resulting operation is confined to a single plane. In practical applications, however, it may be necessary for the tube to operate beyond this plane (i.e., 3D operation). One approach is to rotate the external magnetic field; however, this method is only effective for specific tasks, as changes in the magnetic field direction may interfere with the functioning of other tubes (medical instruments). A more effective solution is to adopt Configuration G of the 1D magnetization reprogramming. This configuration enables 3D deformation under a uniaxial constant magnetic field (movie S16). By combining this configuration with the magnetic neutralization feature, it becomes possible to achieve both diverse 3D deformation modes and non-deformation states under a constant magnetic field (Fig. S14, movie S1). This capability makes it particularly well-suited for realizing 3D coordinated multi-instrument operations.

Comment 5: I suggest the authors provide a clear definition of the term “magnetic unit,” which is used frequently throughout the manuscript. For example, on page 4, line 109, the authors mention “even an infinite number of magnetic units” for Figure 1B(i), which is confusing.

Response:

Thanks for this helpful reminder. In the revised manuscript, we included a definition of the term “magnetic unit” and clarified several other terms.

The added or revised descriptions are in lines 107-121 as:

This study introduces real-time, in-situ magnetization reprogramming, which modifies the relative positions of magnetic units within a soft robot by dynamically and locally manipulating the magnetic unit carriers (Fig. 1A). The variation in magnetic unit positions enables the real-time and in-situ reprogramming of the magnetization profile of soft robots. “A magnetic unit” is defined as a collection of magnetic material within a continuous region that exhibits uniform magnetization direction. This unit is independent of its volume size and is defined solely by the magnetization orientation. For example, in Figs. 1B (ii) and 2A(iv), the black region at the end of the tube, which has a downward magnetization direction, constitutes one magnetic unit (diameter: 1.9 mm and length: 2 mm). In contrast, in Fig. 1B(i), the tube displays a helical magnetization profile where magnetic particles along its axis have varying magnetization directions, with each particle representing a separate magnetic unit (diameter: 0.05 mm). “Magnetic unit carriers” refer to components in soft robots that are capable of transporting magnetic units. They can take the form of nested tubes (Fig.

1A, Fig. 2A(i)) or built-in rods (Fig. 2B(i), Fig. S24), and they undergo movement under the influence of external forces generated by actuators, such as motors, pneumatic, hydraulic, or piezoelectric devices, or by other stimuli.

Comment 6: While the manuscript demonstrates the experimental setup for magnetizing the samples and applying magnetic fields, it lacks details on controlling the relative motion of the nested tubes or rods. Are they manipulated manually, or is there a motorized control system involved?

Response:

This study primarily focuses on reprogramming magnetization profiles and the resulting deformations under magnetic fields. Therefore, in the original submission, we did not elaborate on the design of actuation devices or the implementation of motion control for the tube. The relative movement of the tube can be achieved by various means. To maintain the generality of our proposed method, we intentionally did not restrict it to any specific actuation strategy—since multiple approaches can be applied, including motors, tendons, pneumatic devices, and other types of actuation methods. For example, we only briefly discussed in the “Discussion” section that appropriate stimuli could be used at small scales as alternative actuation methods.

In this study, different control strategies were employed for the tube depending on the specific experiment. For instance, in the demonstrations of deformation under different configurations and in certain operations of Application I, we designed a mechanical control setup using Manipulator A and Manipulator B to independently control the nested tubes. Manipulator A is driven by a *stepper motor* coupled with a lead screw, allowing forward and backward motion of Tube A. Manipulator B is equipped with two *stepper motors* and lead screws, enabling forward/backward movement as well as rotation of Tube B (see Fig. S86, also shown below). The rotational motion is introduced to reduce resistance during relative movement between the tubes (refer to Section 1.2.1 of Note S9, Note S6, Fig. S88B). In Application II, due to the involvement of multiple small-scale tubes, we adopted *manual control* of the inner tube’s relative motion to simplify the experimental setup.

Fig. S86. Schematic diagram and photograph of the tube operating mechanism. (A) Schematic diagram. (B) Photograph.

In addition, we have added the relevant description in lines 224-228 of the revised manuscript as:

Our method does not impose specific constraints on the external forces used to actuate relative motion among magnetic unit carriers. These forces can originate from a wide range of actuators, such as motors (Fig. S86), pneumatic, hydraulic, or piezoelectric devices, or from any form of motion-generating stimulus.

Comment 7: Two important recent works on magnetic continuum robots for contact-free navigation—*Nat. Commun.* 15, 3759 (2024) and *Adv. Funct. Mater.* 2024, 2412543—have not been properly cited. Including these references would strengthen the manuscript’s context within the field.

Response:

Thanks for suggesting these relevant references. In the revised manuscript, we cited both papers and added relevant discussions in lines 275-281. The added discussions are provided as:

In addition, a technique that combines magnets with low-melting-point alloys has been developed (62, 63). This approach uses magnetic forces to deform the tube and leverages the alloy’s solidification and melting processes to maintain or release the tube’s shape. The technique shows the potential for reducing vascular damage. However, heating and cooling within a sealed chamber is time-consuming, which precludes real-time operation and raises concerns about excessive heat accumulation.

Comment 8: In Figure 3B, there are two sections labeled (ii). One of them should be labeled (iii) to maintain proper sequencing.

Response:

Thanks, the necessary corrections have been made in the revised version, and we have also carefully checked the remaining figures and texts to ensure overall consistency and clarity.

Referee #2:

Overall Assessment:

Actuators based on concentric tubes with different magnetization directions in each layer are reported, where axial translation of the tubular layers dynamically alters the total magnetization profile, thus reprogramming the response of the tube. Single tubes are envisioned as surgical catheters that can avoid obstacles while being inserted, and multiple tubes are shown working together in a way that could mimic multiple catheters. Arrays of tubes are demonstrated as dynamically tunable cilia that can pump fluids. Forming bundles of multiple tubes or connecting them with non-responsive elastomer layers demonstrates additional capabilities, such as morphing 2D sheets, boxes, and grabbers. This work is elegant, creative, and comprehensive.

Despite these strengths, in my assessment, this work does not rise to the level of novelty and impact that are needed for publication in Nature, though it comes close. The second paragraph of the Introduction describes the challenge and the need to reprogram the magnetization profile, while referencing approaches that reorient magnetic domains. This work reprograms the total magnetization another way, which is innovative but has some significant limitations: The structures are tethered to adjust the positions of the tubes, and reprogramming is accomplished through partial cancellation of the magnetization, creating a tradeoff between programmability and the strength of the response. Reprogramming through mechanical motion of the tubes will restrict scaling to much smaller sizes. There are also practical limitations to the number of layers in a tube, which will ultimately limit the behaviors attainable through reprogramming. For example, the length of the tubes reported in this work is about 50 mm, but the more clinically relevant case of a catheter with a length of 900 mm is given. It may be possible to avoid a couple of obstacles, but there is no clear approach for avoiding many obstacles.

Response:

We sincerely thank the Reviewer for the constructive comments. The Reviewer raised concerns regarding small-scale manipulation, the number of layers, operational length, and traversal of multiple obstacles. To address these limitations, in the revised manuscript, we added detailed design schemes, optimization analysis and discussion, as well as relevant experimental demonstrations. Below, we provide point-by-point responses to each of these issues.

(1) Small-scale manipulation

Regarding the tube manipulation, we provided a detailed analysis in Sections 1-3 of Note S9 in the revised manuscript. Specifically, we established the mechanical criteria for successful manipulation, namely:

$$\begin{cases} F_{cr} > F_{m-total} \\ F_y > F_{m-total} \end{cases}$$

where F_{cr} is the critical buckling load of the tube, $F_{m-total}$ represents the relative motion resistance between tubes, and F_y is the yield force of the tube. We also proposed a comprehensive methodology for the design, parameter tuning, and performance optimization under various conditions (Sections 1-3, Note S9). This analysis provides a generalized framework for tube motion and is applicable to small-scale designs as well. The primary distinction at small scales lies in the composition of the relative motion resistance $F_{m-total}$. At small scales, van der Waals forces and electrostatic interactions become more significant; however, these factors merely contribute to the composition of $F_{m-total}$ and do not alter the overall analytical framework. Therefore, we believe that by following the proposed design and tuning strategy, it can meet the mechanical conditions required for tube operation even at small scales, thereby enabling effective small-scale manipulation.

To demonstrate the feasibility of manipulation at small scales, we attempted to fabricate tubes with very small diameters. In our previous demonstrations, the smallest tube had an outer diameter of 1.4 mm and was fabricated using a dip-coating method. In our new attempt, we used copper wires with a diameter of 0.08 mm as the core to fabricate the small-scale tubes. However, we found that the dip-coating method was not suitable for producing tubes at such small scales. We gradually increased the diameter of the copper wire, and successful tube fabrication was only achieved when the wire diameter exceeded 1 mm. We attribute this limitation to the dominance of surface tension at small scales and the wettability of the wire. Specifically, at small-scale dimensions, surface tension causes the liquid PDMS to form spherical droplets rather than uniformly coating the wire surface. As shown in Fig. S87A (also shown below) and movie S21, when attempting to fabricate a tube using a 0.5 mm diameter copper wire, the red PDMS gradually aggregated into a spherical shape. We tested various methods to reduce the surface tension of PDMS, but none were successful. Therefore, we adopted an alternative approach by using commercially available miniature tubings (06420-01, Cole-Parmer, US). We fabricated the tube using a combined process of heating, melting, and stretching (see detailed parameters in Fig. S87). The resulting outer tube had an outer diameter of 0.7 mm. We set this tube to Configuration B in the 1D magnetization reprogramming setup and demonstrated its ability to achieve various deformations under a magnetic field (movie S20). Due to the high elastic modulus of the material used in fabrication, a relatively strong magnetic field (250 mT) was applied to induce bending. To confirm that the observed bending was not caused by an internal rod, we also conducted control experiments, which ruled out the influence of the inner rod.

Fig. S87A. Illustration of problems encountered in fabricating miniature tubes using the dip-coating method.

These experiments demonstrate that the proposed method is applicable at sub-millimeter scales. Due to current limitations in tube fabrication, we are not yet able to demonstrate operation at even smaller scales. However, we firmly believe that as long as the fabricated tubes satisfy the mechanical criteria we have established, small-scale manipulation remains feasible.

We added more discussions of the small-scale manipulation in the revised manuscript in lines 466-471 and lines 687-696 as:

The dimensions of the cilia can be adjusted according to specific operational requirements. The cilia demonstrated above have an outer diameter of 1.4 mm. Smaller-scale cilia can also be controlled using the proposed method. In Note S9, we established the mechanical criteria required to achieve relative motion between tubes. Therefore, regardless of the scale, as long as these mechanical conditions are satisfied, our method can be applied to realize a reprogrammable cilia array.

For miniaturization, the proposed method easily applies at the millimeter and sub-millimeter scale (movie S20). However, further size reduction introduces three main issues: difficulties in fabricating soft robotic components, increased friction between components, and reduced positional accuracy of magnetic units. For instance, in this study, we employed the dip coating method to fabricate tubes. However, when the tube diameter is reduced below 1 mm, this approach becomes unsuitable, as the surface tension of the liquid solution dominates, causing it to form spherical droplets rather than uniformly coating the surface of the mandrel (Fig. S87, movie S21). Therefore, alternative fabrication methods need to be explored to produce tubes at smaller scales.

(2) Number of layers

In our original manuscript, we demonstrated typical configurations (Figs. S1-S25), all of which involved the operation of either two-layer or three-layer tube nesting structures. For designs with two nested tubes, multiple in-plane bending deformations could be achieved (Figs. 3C-3E, movie S4), as well as multiple 3D bending deformations (Fig. 3H, movies S16-S17). In the case of three-layer tube nesting, additional modes of deformation can be achieved beyond those available in the two-

layer configuration (Fig. S13, movie S1). However, the number of layers cannot be increased indefinitely. As the number of layers increases, the resulting changes in the overall stiffness of the nested structure become more pronounced, making deformation control more difficult. The configurations presented in Figs. S1-S25 already cover a broad range of typical manipulation requirements. Nevertheless, in certain scenarios, more complex operations may require multi-layered tube designs. To address this, we propose a zone-based tube nesting design method in the revised manuscript to tackle the challenges associated with multi-layer tube configurations (Section 4.4 in Note S9).

By dividing the nested tubes into separate zones, the proposed method reduces excessive stiffness variation during operation. As illustrated in Fig. S94B (shown below), the tube structure is partitioned into multiple functional zones, with each zone containing a fixed number of tubes. Each tube is programmed to move only within its designated zone. As a result, the maximum stiffness variation within any single zone corresponds to the stiffness of only one tube. For example, in Zone 4 of Fig. S94B, five tubes (Tubes 1-5) are present, but the stiffness variation within this zone is determined solely by Tube 5, as the other four tubes remain stationary within this region. Detailed design principles are provided in Section 4.4 of Note S9.

Fig. S94B. Proposed zone-based tube nesting design method.

To demonstrate the feasibility of the proposed zone-based tube nesting design method, we combined it with a multi-magnetic field control strategy described in Section 4.1.3 of Note S9, in which different zones are actuated by distinct magnetic fields. The specific design and experimental setup are shown in Figs. S83-S84. We fabricated Tube A, Tube B, and a rod. Tube A consists of two layers of tubes, as shown in movie S18. Specifically, the outermost black tube is directly fixed onto the next outermost red tube. However, in Fig. S83, we simplified this arrangement according to its configuration and labeled it as Tube A with only one layer. Therefore, although Fig. S83 depicts a three-layer tube configuration, it is actually composed of four layers. In this design, two zones were defined: one containing Tube B and the rod, and the other containing Tube A, Tube B, and the rod. These zones correspond to the two magnetic field regions

generated by the two Halbach arrays in Fig. S84. This design also aligns with the multi-bending deformation strategy discussed in Section 4.2 of Note S9. In the Halbach array located near the operation end, Tube A is directly actuated by the magnetic field and undergoes deformation, thereby driving the deformation of the nested Tube B and the rod. In the Halbach array farther from the operation end, Tube A is no longer present. In this zone, only Tube B and the rod remain, and their relative movement enables the generation of various types of complex 3D deformations. The detailed operation process is presented in movie S18.

Although our validation experiments did not involve tube structures with more layers, they provided preliminary evidence supporting the feasibility of the proposed zone-based tube nesting design method for addressing the challenges associated with multi-layer tube nesting. We believe that this method is also applicable to configurations with a greater number of nested tubes.

(3) Operational length

For the operational length, we presented detailed schemes for long-range operations in Section 4.1 of Note S9 in the revised manuscript. This section includes detailed design schemes, parameter analyses, and optimization strategies for various application scenarios. We subdivided the operating region into three distinct zones (pages 1-2 of Note S9): *Magnetic Interaction Zone* (where the magnetic field is present), *Actuation Zone* (where the tube is actively driven to achieve relative motions), and *Feeding Zone* (connects the Magnetic Interaction Zone and Actuation Zone). Consequently, the total length of the tube that can be operated is given by the sum of the tube lengths within the Magnetic Interaction Zone and the Feeding Zone.

Based on the aforementioned zone division, we categorized the operations into three types for clarity of explanation: long-range operation with a large Magnetic Interaction Length, long-range operation with a large Feeding Length, and long-range operation with a large Magnetic Interaction Length and large Feeding Length, where the third type represents a combination of the first two. Regarding the long-range operation with a large Magnetic Interaction Length, we demonstrated that a tube with a length of 305 mm can be manipulated while achieving multiple bending deformations (Figs. S83-S84, movie S18). This length can be further increased with enhanced stiffness of tubes. For the long-range operation with a large Feeding Length, we integrated commercial catheters for manipulation and demonstrated an achievable operational length of at least 1240 mm (Fig. S85, movie S19). Since there is no magnetic field in the Feeding Zone in this type of operation, the tube does not undergo deformation in this region. Therefore, we employed a commercially available catheter to replace the high-stiffness tube in the Feeding Zone. We believe that the aforementioned operational distances satisfy most application requirements. Furthermore, if necessary, longer distances can be achieved by summing the Magnetic Interaction Length and the Feeding Length.

Additionally, in Section 4.1 of Note S9, we proposed a segmented design method and a multi-magnetic field control method for addressing design challenges associated with

long-range operations. Due to the extensive details involved in these methods, we refrain from elaborating further here; please refer to Section 4.1 of Note S9 for a detailed discussion.

To give more descriptions of the operational length and other considerations, we added more text in lines 342-358 as:

To validate the capability of the proposed method for achieving 3D contact-free object navigation, we fabricated two different 3D objects to avoid contact (Figs. S80-S81). By designing different magnetization profiles, we achieved contact-free navigation through these 3D objects under a single magnetic field (Fig. 3H, Figs. S79 and S82, movie S17). In future practical applications, there will likely be increased demands on the operational length, the number of bends, the curvature, and the number of layers. To address these specific challenges, we proposed four new methods—namely, a segmented design method, a multi-magnetic field control method, a multi-bend deformation design method, and a zone-based tube nesting design method—targeting long-range operation, multi-bend deformation, high-curvature deformation, and multi-nested tube operation, respectively. For example, regarding the long-range operation, we demonstrated that a tube with a length of 305 mm could be manipulated while achieving multiple bending deformations (Figs. S83-S84, movie S18). This length can be further increased with enhanced stiffness of tubes. We integrated commercial catheters for manipulation, demonstrating an achievable operational length of at least 1240 mm (Fig. S85, movie S19). We believe that the aforementioned operational distances essentially satisfy most application requirements. The detailed designs, tuning strategies, and related discussions of these methods are presented in Section 4 of Note S9.

(4) Traversal of multiple obstacles

In our original manuscript, we demonstrated the capability to traverse one or two obstacles. To further illustrate the performance of our method, additional demonstrations of obstacle traversal were included in the revised manuscript. Using a vibrating sample magnetometer (VSM) (EZ7, MicroSense, USA) to generate a uniaxial magnetic field, we demonstrated 3D navigation across both one and two obstacles (Figs. S79-S82, movie S17).

Achieving traversal over a greater number of and more complex obstacles requires a larger magnetic field region and more complex magnetization profiles. However, generating such large-scale magnetic fields is constrained by fabrication, cost, and other practical limitations. To address this, we proposed a multi-magnetic field control method, which utilizes multiple small-scale magnetic fields as an alternative approach (Section 4.1.3 in Note S9). In addition, to simplify the design of magnetization profiles, we introduced a multi-bend deformation design method (Section 4.2 in Note S9). Since we currently do not have appropriate electromagnetic coils for implementing the multi-magnetic field control method, we fabricated two Halbach arrays to substitute for the large-area magnetic field (Fig. S84). Accordingly, we designed a customized tube

configuration based on the multi-bend deformation design method (Fig. S83). In movie S18, the tube demonstrates planar bending under the first magnetic field (similar to 2D obstacle crossing shown in movies S4-S5). Under the second magnetic field, the tube performs 3D navigation (similar to the 3D obstacle traversal in movie S17), and further exhibits the capability of switching between different operating modes. With the addition of more magnetic fields, the system is capable of performing a greater number of bends, thereby enabling it to traverse more obstacles. Since we were unable to control the magnetic field strength of the Halbach array, the operation in movie S18 was implemented with the following two limitations.

1) The tube could not achieve smooth deformation as it did in the VSM setup. Because the magnetic field generated by the Halbach array was constantly present and could not be gradually ramped up, the tube began to bend even before fully entering the magnetic field region.

2) We were unable to demonstrate real-time obstacle avoidance. In the VSM system, the tube was shown to gradually bypass obstacles without contact, enabled by the ability to incrementally increase and decrease the magnetic field strength. However, because the Halbach array does not allow for field strength modulation, we could not demonstrate how the tube navigated around obstacles in real time. As an alternative, we only presented a proof-of-concept demonstration showing that the tube can achieve multiple bending deformations and traverse more obstacles (without actual obstacles being placed in the demonstration) in movie S18. We believe that if the magnetic field strength of the Halbach array is tunable, it will enable multiple deformation modes and complex obstacle traversal.

To address the issue of magnetic field control, we have proposed specific solutions as a continuation of this research (Section 4.2.2 in Note S9). The first solution involves designing a mechanism to adjust the diameter of the circular arrangement of permanent magnets in the Halbach array. By changing the distribution circle diameter, the magnetic field strength can be modulated (see Fig. S89 for the relationship between magnetic field strength and distribution diameter). The second solution is to replace the Halbach array with electromagnetic coils to enable tunable fields. Furthermore, to lay the foundation for future research, we conducted simulations on potential interference issues arising from spatial arrangements of multiple magnetic fields (Note S11).

Please note that, in Sections 4.1-4.4 of Note S9, we proposed four new methods—namely, a segmented design method, a multi-magnetic field control method, a multi-bend deformation design method, and a zone-based tube nesting design method—to address specific challenges in long-range operation, multi-bend deformation, high-curvature deformation, and multi-nested tube operation, respectively. In the experimental demonstrations, some designs incorporate multiple methods simultaneously. As a result, certain demonstrations/videos are referenced multiple times, serving as demonstrations or validations for different design strategies.

In addition to addressing the aforementioned issues, our revised manuscript presents corresponding solutions, optimization strategies, and experimental demonstrations for other potential challenges. Our revised manuscript also includes a theoretical proof of the feasibility of 3D manipulation, a theoretical comparison showing that the force-torque space of existing methods is a subset of that enabled by our proposed method (Fig. S70), a methodology for magnetization profile design, analysis of magnetic field interference, and an investigation into factors affecting tube manipulation. Details can be found in Notes S6-S11 and movies S16-S21.

Comment 1: What causes the orange color of these structures?

Response:

We thank the Reviewer for the helpful comment. Our understanding is that the Reviewer expressed concerns about the red or orange coloration of the tubes, sheets, or other structures shown in our figures and videos. As the tubes are made of PDMS, which is transparent after curing, it is difficult to visually capture their shape and deformation. To enhance the visibility of the structures, we added a red pigment (SILC-PIG™, KauPo, DE) to the PDMS during fabrication. This results in the red or orange appearance of the tubes in the images and videos. For other similar structures, such as the 2D sheets or 3D soft structures, which are also fabricated using PDMS or Ecoflex, we similarly added red pigment to improve visibility and maintain a consistent appearance across all samples.

To avoid potential confusion by the Reviewer or future readers, we added a clarification in lines 767-771 as:

In this study, to enhance visibility, a red pigment (SILC-PIG™, KauPo, DE) was added to all 1D, 2D, and 3D samples during fabrication, with the exception of those containing magnetic particles. The samples with magnetic particles remain black due to the inherent color of the magnetic material, which cannot be masked by the pigment.

Comment 2: Why is the magnetometry measurement in Fig. S65a asymmetrical?

Response:

We sincerely thank the Reviewer for the careful examination of our figures. In fact, when processing the data, we did not pay particular attention to the plotted figure. Instead, we directly used the numerical data provided in the file generated by the vibrating sample magnetometer (VSM) (EZ7, MicroSense, USA) to calculate the magnetization—specifically, the data highlighted in red box C of Fig. R1 below. Upon reviewing all our previously measured data, we found that even when the plotted magnetization curves appear highly symmetrical, the remanence values provided in the VSM data files are not perfectly symmetric. We repeated the measurements multiple times and observed the same phenomenon. For example, after fabricating a new sample

and performing another measurement, we obtained a new VSM data file (see Fig. R2 below), in which the remanence values in red boxes A and B are very close but still not identical. Despite multiple rounds of calibration, this slight discrepancy in remanence consistently persisted. We believe it may be caused by small differences in sample positioning. However, this variation appears to have a negligible effect on the calculation of the average remanence.

We selected a newly measured dataset that is nearly perfectly symmetric and recalculated the magnetization. The results were consistent with those obtained from our previous calculations. Therefore, we did not modify the data presented in Fig. S65B. However, we replaced Fig. S65A with a dataset that exhibits better symmetry (the volume of the new sample is 38.2 mm³).

System ID: EV X, SN: XXXXXXXX, Customer: XXXXXXXX, first started on: Monday, November 13, 2023
 Date and time of last calibration: Friday, December 15, 2023 10:56:30
 Operator: System_Admin
 Sample name: xianqiantg_0.5v1
 Please Select Sample Holder and Orientation
 Data filename: c:\vsm-1\System_Admin\data\xianqiantg_05v1\xianqiantg_05v1-Hys-a000-RT.VHD
 Start of measurement: 11:05:21, Friday, December 15, 2023
 Field Angle: -0.00 [deg]
 Total measurement time: 00:18:49

Parameters	Up	Down	Average	Parameter Definition
Hysteresis Loop				Hysteresis Parameters
M at H max emu	355.294E-3	-363.421E-3	359.313E-3	M at the maximum field
PP emu	718.085E-3	-716.421E-3	717.526E-3	Peak to Peak Signal (Max + Min)
Stdev emu	231.385E-3	230.857E-3	235.257E-3	Standard Deviation
Hc Oe	6432.901	-7954.912	4700.907	Coercive field: field at which M/H changes sign
M _s emu	355.204E-3	-363.421E-3	359.313E-3	Saturation Magnetization: maximum M measured
Hc offset Oe	6442.93	2958.91	1741.93	(Hc upward curve + Hc Downward curve)/2
M _r emu	-248.065E-3	129.460E-3	188.763E-3	Remanent Magnetization: M at H=0

A
B
C

MicroSense EasyVSM Software Version EasyVSM 20190206-01

Fig. R1. VSM data file obtained from the previous measurement.

System ID: EV X; SN: XXXXXXXX; Customer: XXXXXXXX; first started on: Monday, November 13, 2023
 Date and time of last calibration: Thursday, March 20, 2025 17:55:06
 Operator: System_Admin
 Sample name: xianqiang-4
 5 mm Quartz Transverse
 Data filename: c:\vsm-IV\System_Admin\data\xianqiang-3\xianqiang-3-Hys-a000-RT.VHD
 Start of measurement: 17:59:00, Thursday, March 20, 2025
 Field Angle: -0.00 [deg]
 Total measurement time: 00:28:08

Parameters				
	Up	Down	Average	Parameter 'definition'
Hysteresis Loop				Hysteresis Parameters
M at H max emu	1.812E+0	-1.857E+0	1.835E+0	M at the maximum field
PP emu	3.654E+0	3.662E+0	3.669E+0	Peak to Peak Signal (Max - Min)
Stdv emu	1.047E+0	1.045E+0	1.280E+0	Standard Deviation
Hc Oe	5001.857	-4943.803	4972.830	Coercive Field: Field at which M/H changes sign
Ms emu	1.812E+0	-1.857E+0	1.835E+0	Saturation Magnetization: maximum M measured
Hc offset Oe	5001.86	-4943.80	29.03	(Hc upward curve + Hc Downward curve)/2
Mr emu	-1.123E+0	1.096E+0	1.109E+0	Remanent Magnetization: M at H=0

A B C

MicroSense EasyVSM Software Version EasyVSM 20190206-01

Fig. R2. VSM data file obtained from the new measurement.

Comment 3: Is there an error in Fig. S66B(vi)? The negative mold shows narrow tips that would be completely filled, but they are missing two panels below in (viii).

Response:

We sincerely thank the Reviewer for the careful review of our figures. After a thorough examination, we found that Fig. S66B(vi) is correct, and the error instead lies in Fig.

S66B(viii), where a drawing mistake occurred. As shown in the figure below, the negative mold contains narrow tips (cylindrical grooves) marked by red circle A, which serve as positioning features for the positive mold. After pouring the solution into the negative mold and inserting the positive mold for assembly, the cylindrical grooves are occupied by the rods of the positive mold (red circle B). Once the material is cured and the positive mold is removed, the ends of the rods should remain free of cured material (red circle C). However, in our original submission, we mistakenly illustrated cured material at the rod ends of the positive mold. We have now corrected Fig. S66B(viii) by modifying the illustration of the rod tips accordingly, as shown in the updated red circle C in the figure below.

Part of Fig. S67. Fabrication process of the tubes.

Comment 4: For avoiding obstacles and coordinated operation of multiple catheters, it seems like the anatomy would need to be known in advance to appropriately magnetize the tubes and plan the motion. Is that correct, or is it possible with this approach to navigate in an unknown environment, based only on imaging near the tip of the catheter?

Response:

We thank the Reviewer for the constructive suggestion, which provides valuable guidance for our future applications. Regarding whether prior knowledge of anatomical structures is necessary, we believe this depends on the complexity of the unknown environment. Our tube is capable of real-time reprogramming of its magnetization profile, and we have demonstrated its ability to achieve a variety of deformations under a constant magnetic field. When combined with tunable magnetic fields, the range of achievable deformations becomes even broader (Fig. S70). When the complexity of the external environment exceeds the range of deformations that can be generated by the tube, prior knowledge of the environment is necessary to design appropriate magnetization profiles and magnetic field patterns. However, for simpler environments, even if unknown in advance, real-time adjustment of the magnetization profile or magnetic field—guided by imaging feedback—can be used to achieve the desired deformation and accomplish the intended task.

Of course, if prior anatomical information is available, we can pre-design the magnetization profile and magnetic field, which would facilitate navigation. For certain

environments—such as specific anatomical structures—the inter-individual variation is often limited. For example, for a particular blood vessel or stomach region, magnetization profiles and magnetic field parameters can be pre-designed based on anatomical data from common individuals. Even though there may be differences across patients, these variations are generally small, and real-time adjustment of the magnetization profile or magnetic field can still allow the tube to adapt effectively.

Furthermore, in Note S8, we theoretically demonstrated that the force–torque space achievable by existing methods is a subset of the force–torque space that can be generated by our proposed method when combined with varying magnetic fields (Fig. S70). Currently, many studies attempt to achieve navigation by integrating existing methods with imaging techniques. In comparison, our method—when combined with imaging—offers greater potential and advantages for navigation tasks.

Referee #3:

Overall Assessment:

Metin Sitti and his colleagues report core-shell structured multiple-tubes using movable magnetic units as core materials. By doing so, real time in-situ reprogramming of magnetization and magnetic actuations were enabled. The research team then demonstrated potential applications in biomedical fields. The demonstration of biomedical applications with controllable tube configurations is impressive due to its potential for practical applications. Conclusions are well-supported by the extensive amount of experimental data for parameter studies with theoretical analysis.

However, I have mixed feelings about reading this manuscript. While the demonstration is neat, basic idea has been shown several years ago. The authors stated that “The real-time in-situ magnetization reprogramming proposed in this paper is achieved by dynamically altering the positions of magnetic units”. Core-shell structure with different position of magnetic cores have been shown in 2020 *Advanced Materials* (*Adv. Mater.* 2020, 32, 2001879) by Zhengzhi Wang. Parameter studies have shown in 2020 *Extreme Mechanics Letters* (*Extreme Mechanics Letters* 38 (2020) 100734). The same author reported reprogramming of magnetic actuation by dynamic re-location of magnetic particles in core-shell structured micropillars in 2021 *ACS Nano* (*ACS Nano* 2021, 15, 4747–4758). The previous study also showed writing and rewriting of area-selective patterned actuation of micropillars (or cilia for terminology in the manuscript). While the submitted manuscript is showing magnetic reprogramming not just location of magnetic particles, this was possible due to cm scale of tubes. The 2021 *ACS Nano* paper showed writing and rewriting of 10 micron-sized pillar arrays. In particular, see their Figure 4, SI Video 10 and SI Video 11 for patterning of micropillar actuations.

Surprisingly, none of the aforementioned previous papers are cited in this manuscript although this manuscript cited 92 references. Hence, if this is not intentional, I believe that the authors are totally unaware of these previous reports. While this manuscript utilizes the core-shell tubes for faster relocation of core tube, this is possible due to the larger tube size. The same concept would not be available for micron scale. The 2021 *ACS Nano* shows relocation of magnetic units within 10-20 s.

Due to the presence of previous reports on the key concepts including smaller scale demonstration, I cannot support for publication of this manuscript in *Nature*. I recommend authors to transfer this manuscript to *Nature Materials* or *Nature Communications*.

Response:

We sincerely thank the Reviewer for carefully reviewing our manuscript and providing valuable comments and relevant references. We have thoroughly read the three references suggested by the Reviewer and carefully compared the differences between

our proposed method and those presented in these references. For clarity, we have labeled the three references as follows:

Ref-1: Hybrid Magnetic Micropillar Arrays for Programmable Actuation, *Advanced Materials*, 2020, 32,2001879.

Ref-2: Heterogeneous magnetic micropillars for regulated bending actuation, *Extreme Mechanics Letters*, 2020, 38, 100734.

Ref-3: Core-Shell Magnetic Micropillars for Reprogrammable Actuation, *ACS, Nano* 2021, 15, 4747-4758.

We found that all three studies share a similar underlying concept, namely achieving varying deformation behaviors in micropillars by generating different concentration distributions of magnetic nanoparticles under applied magnetic fields. However, the three studies differ in their research focus: Ref-1 and Ref-2 investigate magnetization programming techniques, while Ref-3 explores magnetization reprogramming techniques. The method we proposed in our study falls into the category of magnetization reprogramming.

We emphasize that magnetization programming and reprogramming are fundamentally different concepts. Magnetization programming refers to the design and embedding of magnetization profiles into target structures/robots before fabrication. In this approach, the magnetization profile becomes fixed after manufacturing and cannot be altered afterward, limiting the structure/robot to specific operational scenarios. In contrast, magnetization reprogramming allows for modifying the magnetization profiles at any stage—either before or after manufacturing—according to varying operational demands. This capability significantly enhances the functionality and adaptability of the target structures/robots. Ref-3 also clearly distinguishes between these two technologies, stating: “it is highly desirable to achieve both programmable (i.e., the structures can deform in a predesigned manner) and reprogrammable (i.e., the structures can reversibly and recurrently deform into multiple configurations as demanded) actuated deformations...”

Given that Ref-1, Ref-2, and Ref-3 present methods similar to each other, and only Ref-3 aligns with our approach of magnetization reprogramming, we will primarily focus our comparative analysis on Ref-3. The comparative analysis will be presented in terms of the following four aspects: technical strategy, resulting performance, application scope, and impact on the current field.

1. Technical strategy

Ref-3 utilizes an external magnetic field to drive magnetic nanoparticles, positioning these particles at varying locations within micropillar cavities. This positioning then results in different deformation behaviors under an applied magnetic field.

Our proposed method achieves the repositioning of magnetic units directly through the internal movement of soft robotic components themselves, subsequently enabling deformation under external magnetic fields.

The fundamental difference in technical strategy lies in *the method employed to alter the positions of nanoparticles or magnetic units*.

2. Resulting performance

In Ref-3, the repositioning of nanoparticles must first be accomplished using a specialized device (a magnetic needle) before any deformation can occur under external magnetic fields. This procedure is analogous to the methods we cited in our manuscript (in **lines 57-65**), which employ magnetic fields to reorient magnetic domains after heating or phase transitions. Such methods, including that of Ref-3, necessitate additional equipment and time to modify the magnetization profile, thereby limiting real-time operations. Furthermore, the deployment of these additional devices will be restricted by the operational workspace, consequently constraining in-situ operability.

In our proposed method, however, *repositioning is achieved through the relative motion of the soft robot's components, entirely independent of the external magnetic field*. The repositioning operation and magnetic actuation can occur simultaneously, thus enabling real-time magnetization reprogramming. Moreover, as these components are integral parts of the soft robot itself, no additional equipment is required for magnetization reprogramming, which facilitates in-situ operability.

Therefore, the critical difference in resulting performance is that *the method described in Ref-3 does not allow for real-time or in-situ magnetization reprogramming, whereas our proposed method successfully enables both*, clearly aligning with the title of our manuscript.

3. Application scope

In Ref-3, nanoparticles have limited positioning options, confined to either the upper or lower half of micropillar cavities. Consequently, the deformation modes are limited to two distinct types: small deformation micropillars (SDP) and large deformation micropillars (LDP), as shown in **Fig. R3-A** below. These configurations provide only minor variations in micropillar deformation, thus constraining the range of applications. Additionally, during operation, the magnetic field (for micropillar deformation) must be applied within a specific angular range relative to the micropillar axis; otherwise, the magnetic field would not only deform the micropillars but also displace the nanoparticles. Consequently, the magnetic field used to induce micropillar deformation cannot be arbitrarily applied, which significantly limits its range of applications.

[REDACTED]

Fig. R3. Bending mode and bending amplitude. (A) Ref-3 (Fig. 3 in Ref-3).

(B) Our proposed method (Fig. 4 in our manuscript).

Our proposed method, however, allows for *arbitrary positioning of magnetic units*, theoretically enabling an infinite variation of deformations. Furthermore, the control of magnetic units in our proposed method is entirely independent of external magnetic fields, allowing the external magnetic fields to be applied freely.

To facilitate a more appropriate comparison with Ref-3, we selected “Application III- Reprogrammable Cilia Array” from our study, which presents an application scenario similar to that in Ref-3. Within this application, our method enables three distinct types of control: targeted cilia activation, ciliary bending amplitude modulation, and ciliary phase modulation. Each type allows for refined, detailed control. For instance, within ciliary bending amplitude modulation, our approach enables arbitrary adjustments to bending amplitude (see Fig. R3-B), whereas Ref-3 can achieve only two discrete amplitude settings (see Fig. R3-A). Therefore, the method in Ref-3 represents only a special case of the ciliary bending amplitude modulation that our method can achieve. Furthermore, targeted cilia activation and ciliary phase modulation are functionalities that Ref-3 cannot realize.

4. Impact on the current field

The method proposed in Ref-3 closely aligns with previously cited methods in our manuscript (in lines 57-65). Both Ref-3 and these cited methods rely on external magnetic fields to reposition magnetic domains, placing them within a similar

technological framework. By contrast, our proposed approach is entirely independent of external magnetic fields, representing a significant technological advancement beyond existing frameworks. This breakthrough not only facilitates unprecedented modes and varieties of deformation but also enables real-time and in-situ operation. Consequently, our method substantially expands the applicability and versatility of magnetic actuation technologies, opening new avenues for development.

In summary, the methods presented in the literature provided by the Reviewer differ distinctly from our proposed method concerning technical strategy, resulting performance, application scope, and impact on the current field. We hope these clarifications highlight the contributions of our research and its distinction from existing studies.

Furthermore, since Ref-3 focuses on magnetization reprogramming, we regret omitting its citation. In our revised manuscript, we cited and discussed Ref-3 in lines 65-71 and added discussions as below:

Meanwhile, another method employs a viscous liquid to localize magnetic nanoparticles, eliminating the previously mentioned heating and cooling processes but introducing several drawbacks, such as being limited to only two types of magnetization profiles and restrictions on the orientation of the driving magnetic field (35). Moreover, since this approach still requires additional equipment to drive the slow motion of nanoparticles in a highly viscous medium, it is incapable of achieving real-time, in-situ magnetization reprogramming.

Response to Reviewer Comments

We thank all reviewers for their insightful comments and constructive feedback. Below, we replied to comments point-by-point, where the reviewer comments are in blue, our replies are in black, and changes in the manuscript are highlighted in yellow.

Referee #1:

Comment:

The resubmitted manuscript has well addressed the concerns raised in my previous review. I am satisfied with the revisions and consider the work suitable for publication in Nature.

Response:

Thank you very much for your positive assessment. We are delighted that our revisions have fully addressed your previous concerns and that you consider the manuscript suitable for publication. We deeply appreciate your time and valuable feedback throughout the review process.

Referee #2:

Comment:

I appreciate the comprehensive responses to the reviews. The authors have addressed my concerns and make a persuasive case for the suitability of this work in Nature. The mechanism of dynamic reprogramming is distinct from other work and impactful for applications. The additional demonstrations added in the revision highlight this potential.

Response:

Thank you very much for your positive comments. We are pleased that our amendments have resolved your earlier concerns. Your thorough evaluation and professional guidance throughout this review have been immensely appreciated.

Referee #3:

All Comments:

The main claim of this manuscript is gradient induced programming of local deformation and reprogramming of magnetic actuation modes. As I pointed out from last review, there are three very relevant but uncited papers.

Ref-1: Hybrid Magnetic Micropillar Arrays for Programmable Actuation, *Advanced Materials*, 2020, 32,2001879.

Ref-2: Heterogeneous magnetic micropillars for regulated bending actuation, *Extreme Mechanics Letters*, 2020, 38, 100734.

Ref-3: Core-Shell Magnetic Micropillars for Reprogrammable Actuation, *ACS, Nano* 2021, 15, 4747-4758.

Although Ref-1 and Ref-2 are not dealing with magnetic reprogramming, varying deformation behaviors in micropillars were generated by moving magnetic nanoparticles under applied magnetic fields.

“We found that all three studies share a similar underlying concept, namely achieving varying deformation behaviors in micropillars by generating different concentration distributions of magnetic nanoparticles under applied magnetic fields. However, the three studies differ in their research focus: Ref-1 and Ref-2 investigate magnetization programming techniques, while Ref-3 explores magnetization reprogramming techniques.”

Hence, the authors admit that the novelty of this manuscript is not about varying deformation behaviors based on mechanical stiffness gradient to program local deformation, but about magnetization reprogramming. The method we proposed in our study falls into the category of magnetization reprogramming.

“Given that Ref-1, Ref-2, and Ref-3 present methods similar to each other, and only Ref3 aligns with our approach of magnetization reprogramming, we will primarily focus our comparative analysis on Ref-3. The comparative analysis will be presented in terms of the following four aspects: technical strategy, resulting performance, application scope, and impact on the current field.”

Also, authors admit that the magnetization reprogramming has been reported before.

“Our proposed method achieves the repositioning of magnetic units directly through the internal movement of soft robotic components themselves, subsequently enabling deformation under external magnetic fields.”

The Ref-3 repositioned the location of magnetic nanoparticles within micropillars. Meanwhile, this manuscript utilized large millimeter scale tubes and that's why the control of internal movement is allowed. If authors can achieve the same thing with micropillars, I will definitely say yes to this manuscript for Nature. However, internal movement of magnetic nanoparticles within micropillar is reported in 2021 ACS Nano. The 2021 ACS Nano reported magnetization reprogramming and resultant change in actuation behaviors due to stiffness gradient. The concept-wise, the idea is quite similar and the previous paper was published 4 years ago. In addition, the size of actuator was much smaller. In case of micron-scale tubes, it will be significantly more difficult to achieve the same idea due to the fabrication difficulty, difficult control, large friction, and adhesion between tubes after actuation.

“In Ref-3, the repositioning of nanoparticles must first be accomplished using a specialized device (a magnetic needle) before any deformation can occur under external magnetic fields”

This is to remotely control the small scale components (nanoparticles). Macro-scale parts would not need remote control.

“In our proposed method, however, repositioning is achieved through the relative motion of the soft robot’s components, entirely independent of the external magnetic field. The repositioning operation and magnetic actuation can occur simultaneously, thus enabling real-time magnetization reprogramming.”

I do agree with the important novelty and first demonstration of this new concept in macroscale. Authors showed beautiful demonstrations and extensively investigated the phenomena. Other than the novelty issue, every component of the manuscript is at top-notch. However, I still don’t agree with the publication of this manuscript in Nature due to the previous publication (2021 ACS Nano) for important concept on magnetization reprogramming and mechanical stiffness-induced change in deformation of micron-scale actuators with control of position of nanoscale components. Hence, I again recommend transferring this manuscript to Nature Materials or Nature Communications.

Response:

We are grateful to the reviewer for the careful re-evaluation of our manuscript and for providing these comments. To ensure a thorough, point-by-point response, we organized the feedback into the following four items.

Comment 1:

Hence, the authors admit that the novelty of this manuscript is not about varying deformation behaviors based on mechanical stiffness gradient to program local deformation, but about magnetization reprogramming. The method we proposed in our study falls into the category of magnetization reprogramming.

Response:

Yes. The novelty of our manuscript resides in magnetization reprogramming—more specifically, in real-time, in-situ magnetization reprogramming, as reflected in our title, “Real-Time In-Situ Magnetization Reprogramming for Soft Robotics.” We do not claim to be the first to introduce magnetization reprogramming; prior work, including the reviewer-cited Ref. 3, has already explored this concept.

When applying our proposed method for magnetization reprogramming, we observe that different deformation states induce mechanical stiffness gradients, which in turn influence actuation behavior. This intrinsic property can be harnessed to facilitate diverse deformation modes and to assist, for example, in the coordinated operation of multiple instruments, as demonstrated in the “Coordinated multi-instrument operation”

section. While we do not present this effect as our primary innovation, it naturally emerges from our approach and offers practical utility during implementation.

Indeed, as we stated in our previous round of reviewer responses, the reviewer-cited Reference 3 (Ref-3) pertains directly to magnetization reprogramming.

Comment 2:

Also, authors admit that the magnetization reprogramming has been reported before.

Response:

Yes, magnetization reprogramming has been reported previously. We provided a detailed analysis and discussion of current research on this topic in Supplementary Note 1 (or the “Introduction” section in the prior manuscript version).

Comment 3:

The Ref-3 repositioned the location of magnetic nanoparticles within micropillars. Meanwhile, this manuscript utilized large millimeter scale tubes and that’s why the control of internal movement is allowed. If authors can achieve the same thing with micropillars, I will definitely say yes to this manuscript for Nature. However, internal movement of magnetic nanoparticles within micropillar is reported in 2021 ACS Nano. The 2021 ACS Nano reported magnetization reprogramming and resultant change in actuation behaviors due to stiffness gradient. The concept-wise, the idea is quite similar and the previous paper was published 4 years ago. In addition, the size of actuator was much smaller. In case of micron-scale tubes, it will be significantly more difficult to achieve the same idea due to the fabrication difficulty, difficult control, large friction, and adhesion between tubes after actuation.

I do agree with the important novelty and first demonstration of this new concept in macroscale. Authors showed beautiful demonstrations and extensively investigated the phenomena. Other than the novelty issue, every component of the manuscript is at top-notch. However, I still don’t agree with the publication of this manuscript in Nature due to the previous publication (2021 ACS Nano) for important concept on magnetization reprogramming and mechanical stiffness-induced change in deformation of micron-scale actuators with control of position of nanoscale components. Hence, I again recommend transferring this manuscript to Nature Materials or Nature Communications.

Response:

Thank you for the reviewer’s comment. However, we respectfully disagree with the assertion that our work follows the same concept as Ref. 3. In Ref. 3, reprogramming is accomplished by applying an external magnetic field that drives all magnetic nanoparticles within the field region to either the proximal or distal tip of the micropillars, thereby producing two predetermined magnetization states. By contrast,

our approach leverages the internal motion of soft-robotic components to reposition individual—or any desired subset of—magnetic units to arbitrary locations, enabling fully reprogrammable magnetization profiles. Thus, the underlying concepts and the effects they produce are fundamentally distinct.

Furthermore, as detailed in our previous round of reviewer responses, we conducted a side-by-side comparison of the two studies across four key dimensions—technical strategy, resulting performance, application scope, and impact on the field. Our approach demonstrates pronounced advantages at the millimeter scale while remaining amenable to smaller-scale implementation. In the “Discussion” section, we address the challenges inherent to miniaturization and discuss the feasibility of thermal, optical, or chemical actuation modalities for operation at reduced dimensions. **Supplementary Note 9** further specifies the mechanical requirements and optimization strategies for tube control, showing that—once these criteria are satisfied—small-scale tube manipulation is achievable. We believe our proposed method will offer a valuable reference for advancing magnetically controlled soft robotics.

Referee #4:

Overall Assessment:

The manuscript entitled “Real-Time In-Situ Magnetization Reprogramming for Soft Robotics” reports a method for recombination of magnetic units to achieve varied magnetization profiles, thus enabling in-situ shape manipulation of magnetic soft robots without changing external magnetic field. Comprehensive application scenarios are demonstrated and well-supported by experimental results.

Overall, the proposed method endows the magnetic soft robot with impressive deformation capabilities under simple magnetic field control and the manuscript is also well-organized. However, some inherent limitations listed as follows constrain the novelty, flexibility, and application potential of the proposed method, which makes the work not yet meet the standards required for publication in Nature.

Response:

We thank the reviewer for evaluating our manuscript and providing insightful comments. Below, we provide a point-by-point response to each of the suggestions.

Comment 1:

(1) Unlike other magnetization reprogramming methods, the method proposed in this manuscript does not allow for arbitrary control of the magnetization direction at a specific point/region. Instead, the magnetization can only be adjusted to a limited extent by combining multiple magnetized units. As such, this method might be more accurately described as magnetization recombination or redistribution, rather than fully magnetization reprogramming. (2) Once the number and magnetization directions of

the magnetic units in a given configuration are settled, all possible deformation patterns under a constant magnetic field are essentially determined.

Response:

We thank the reviewer for these valuable comments.

(1) For the definition of reprogramming, the reviewer’s interpretation—that magnetization reprogramming should allow arbitrary control of the magnetization direction at a specific point or region—is indeed valid for many existing studies, particularly those based on phase change of the base materials or Curie-temperature-driven domain reorientation. However, we view “reprogramming” in contrast to “programming” to encompass any method that can reproducibly alter the magnetization profile or shape. For example, [R-1] indicates that the term “programmable” is used to describe structures capable of deforming in a predesigned manner, whereas “reprogrammable” denotes structures that can reversibly and recurrently deform into multiple configurations on demand. Furthermore, the work in [R-1] achieved only two discrete magnetization profiles, yet it was still termed the process “reprogramming.” Therefore, we believe it is appropriate to term our technique “magnetization reprogramming.”

(2) While our proposed method does not permit arbitrary reorientation of each magnetic unit’s magnetization direction, it can adjust the magnetization direction within a defined range and freely reposition the magnetic units. Therefore, similar to existing methods, our proposed method can also produce a diverse set of deformations. As shown in Supplementary Figs. 2–26, a single configuration can produce numerous distinct shape changes. In contrast, existing techniques mentioned by the reviewer allow unrestricted modulation of magnetization direction but cannot alter unit positions. So, while certain deformation modes remain unattainable by existing methods, our approach can realize them. Beyond sheer mode diversity, our technique emphasizes real-time, in situ operation, enabling capabilities unavailable to earlier methods—such as navigating around objects without undesired contact, reprogramming cilia arrays, and managing multiple instruments cooperatively or independently under the same magnetic field.

[R-1] K. Ni, Q. Peng, E. Gao, K. Wang, Q. Shao, H. Huang, L. Xue, Z. Wang, Core-Shell Magnetic Micropillars for Reprogrammable Actuation. *ACS Nano* **15**, 4747–4758 (2021).

Comment 2:

In comparison to other magnetization reprogramming methods, the method presented in this work may compromise two key advantages typically associated with magnetic soft robots: untethered control and ease of miniaturization. This could substantially limit the practical applicability of robots based on this approach. The reconfiguration of magnetic units appears to require the use of many tethered actuators, which may result in a relatively bulky system. In this context, it may be worth considering whether magnetic actuation remains the most appropriate choice, especially when alternative

tethered actuation strategies could potentially offer higher precision and more degrees of freedom.

Response:

We thank the reviewers for their insightful comments.

We contend that whether one compromises the two key advantages of magnetic soft robots—untethered control and ease of miniaturization—depends on the specific application. For example, the magnetic soft robots described in [R-2] and [R-3] employ tethered configurations yet still deliver excellent performance. Likewise, in the four application scenarios we present, although we also use tethered setups and millimeter-scale tubes (i.e., not exceptionally small), we achieve functionalities unattainable by existing approaches.

For miniaturization, our proposed method easily applies at the millimeter and sub-millimeter scale. In Supplementary Note 9, we further specify the mechanical requirements and optimization strategies for tube control, showing that—once these criteria are satisfied—small-scale tube manipulation is achievable. Further considerations regarding miniaturization are addressed in the “Discussion” section of the manuscript (see lines 502–517). We also demonstrated the feasibility of implementing our proposed method at a smaller scale in Supplementary Video 20. For achieving the applications demonstrated in our manuscript—such as contact-free object navigation, reprogrammable cilia arrays, and coordinated multi-instrument operation—alternative tethered actuation strategies beyond magnetic actuation currently cannot perform effectively. We are confident that, although our method presents greater miniaturization challenges than existing approaches, these obstacles can be overcome, and the benefits obtained are substantial.

Regarding the number of actuators, we discussed this in detail in our response to Comment 3 and also included it in the “Discussion” section of the manuscript (see lines 493–502).

[R-2] Y. Kim, G. A. Parada, S. Liu, X. Zhao, Ferromagnetic soft continuum robots. *Science Robotics* **4**, eaax 7329 (2019).

[R-3] Y. Kim *et al.*, Telerobotic neurovascular interventions with magnetic manipulation. *Science Robotics* **7**, eabg 9907(2022).

Comment 3:

The design space of the proposed method is directly tied to the number of magnetic units, which in turn determines the number of additional actuators required. Moreover, achieving higher degrees of freedom during the recombination of magnetic units would necessitate a significant increase in the number of actuators. For 1D configurations, the actuator count may still be manageable—as demonstrated by the authors using multiple motors. However, when higher resolution is needed, or when the configuration becomes

2D or 3D, the number of actuators required could grow substantially, leading to a highly complex actuation and control system. This would greatly limit the practical application of the method. For instance, in the manuscript, all reconfigurations beyond 1D are carried out manually.

Response:

We thank the reviewer for these insightful comments.

The number of magnetic units does not directly dictate the number of actuators; rather, the actuator count is determined by the number of magnetic unit carriers. For instance, in the 1D magnetization reprogramming—regardless of whether a configuration contains one, two, three, or numerous magnetic units (Configurations A–E and G; Supplementary Figs. 2–13 and 15)—only a single actuator is required, yet highly complex deformations are achieved. Here, “actuator” refers exclusively to the device used to modify the magnetization profile, and does not include any additional actuators employed for other tube-control functions. For example, in the “Contact-Free Object Navigation” section (Supplementary Fig. 87), three actuators are deployed, but only one serves to reprogram the magnetization profile. The same principle applies to 2D and 3D magnetization reprogramming. Moreover, because 1D reprogramming underpins both 2D and 3D implementations, we concentrated our analysis and applications on the 1D case. Accordingly, we developed a dedicated motor-based control apparatus for 1D magnetization reprogramming, rather than designing separate control systems for 2D and 3D magnetization reprogramming.

In achieving complex deformations, we indeed confront the reviewer’s concern regarding actuator count: more actuators entail greater actuation and control complexity, as well as higher costs. One effective strategy to minimize the number of actuators is to employ configurations with more sophisticated, preprogrammed magnetization profiles. For instance, Configuration G in the 1D magnetization reprogramming (Supplementary Fig. 15) realizes linear, helical, and other intricate deformations with a single actuator. A detailed methodology for designing magnetization profiles is provided in Supplementary Note 10. While reducing the actuator count simplifies the control architecture, it also increases the difficulty of both design and fabrication of soft robots, necessitating a careful trade-off between the complexity of the magnetization profile and the number of actuators. We have added the following discussion to the “Discussion” section of the revised manuscript:

Thirdly, complex deformations may require a greater number of magnetic unit carriers, which in turn necessitates additional actuators for control. An increase in actuator count compounds the complexity of the actuation and control systems and raises costs. One effective strategy to reduce actuator number is to employ configurations with more sophisticated, preprogrammed magnetization profiles (e.g., Configuration G in 1D magnetization reprogramming). We provide a detailed methodology for designing magnetization profiles in Supplementary Note 10. Although fewer actuators simplify the control architecture, complex preprogrammed magnetization profiles increase both

design and fabrication challenges, necessitating a careful balance between profile complexity and actuator quantity.

Comment 4:

The fabrication and inverse design of the proposed robot are also non-trivial. Considering that multiple magnetic units need to be integrated, manufacturing becomes particularly challenging when the robot size is small. Furthermore, in complex and unpredictable environments, it is difficult to design a single configuration that can adapt to diverse scenarios. This poses significant challenges in determining the appropriate number of magnetic units, their magnetization directions, and the corresponding actuation modes.

Response:

We thank the reviewer for their constructive comments.

(1) Fabrication at small scales

Our proposed method is readily applicable at the millimeter and submillimeter scales. The challenges associated with further miniaturization are discussed in detail in the “Discussion” section of the manuscript (see lines 502–517). Additionally, Supplementary Note 9 specifies the mechanical requirements and optimization strategies for manipulation, demonstrating that—once these criteria are met—small-scale robot manipulation is achievable. Besides, we demonstrated the feasibility of implementing our proposed method at a smaller scale in Supplementary Video 20. Analogous to our response to Comment 2, we are confident that, although our method presents greater miniaturization challenges than existing approaches, these obstacles can be overcome, and the benefits obtained are substantial.

(2) Inverse design

We provided a range of relevant configurations and their corresponding deformation behaviors in Supplementary Figs. 2–26, which we believe will accommodate the vast majority of applications. For instance, the four applications presented in the main text are all based on configurations from Supplementary Figs. 2–26. More complex deformation profiles or specialized use cases will require tailored designs; we describe a detailed magnetization profile design procedure in Supplementary Note 10. Furthermore, Supplementary Note 9 provides comprehensive methods for specific design and debugging. For specific application scenarios—long-range operation, multi-bend deformation, high-curvature deformation, and multi-nested tube operation—we propose four novel methods: a segmented design method, a multi-magnetic field control method, a multi-bend deformation design method, and a zone-based tube nesting design method. We believe that these design methodologies—broadly applicable to the vast majority of operations—will enable the development of configurations that meet precise application requirements.

(3) Complex and unpredictable environments

The applicability of our approach in complex and unpredictable environments depends on the degree of environmental complexity. Our method enables real-time reprogramming of magnetization profiles and has demonstrated a variety of complex deformations under a constant magnetic field (Supplementary Figs. 2–26). When paired with a tunable magnetic field, an even broader range of deformations becomes possible (Supplementary Note 8, Supplementary Fig. 71). Therefore, for most unpredictable environments, real-time adjustment of either the magnetization profile and the applied magnetic field should suffice to produce the desired deformation and accomplish the intended task.

However, if the complexity of the external environment exceeds the deformation range achievable by the proposed method, prior characterization of the environment is necessary to design appropriate magnetization profiles and magnetic field patterns. We consider this requirement reasonable, as any specific design inherently has a finite operating envelope and cannot accommodate every possible condition.